



# A regional $p$CO$_2$ climatology of the Baltic Sea from in situ $p$CO$_2$ observations and a model-based extrapolation approach

Henry C. Bittig[1], Erik Jacobs[1], Thomas Neumann[1], and Gregor Rehder[1]

[1]Leibniz Institute for Baltic Sea Research Warnemünde (IOW), Rostock-Warnemünde, Germany

**Correspondence:** Henry Bittig (henry.bittig@io-warnemuende.de)

**Abstract.** Ocean surface $p$CO$_2$ estimates are of great interest for the calculation of air-sea CO$_2$ fluxes, oceanic uptake of anthropogenic CO$_2$, and eventually the Global Carbon Budget. They are accessible from direct observations, which are discrete in space and time and thus always sparse, or from biogeochemical models, which only approximate reality. Here, a combined method for the extrapolation of $p$CO$_2$ observations is presented that uses (1) model-based patterns of variability from an EOF analysis of variability with (2) observational data to constrain EOF pattern amplitudes in (3) an ensemble approach, which locally adjusts the spatial scale of the mapping to the density of the observations. Thus, data-constrained, gap- and discontinuity-free mapped fields including local error estimates are obtained without the need for or dependence on ancillary data (like, e.g., satellite sea surface temperature maps). This extrapolation approach is generic in that it can be applied to any oceanic or coastal region covered by a suitable model and observations. It is used here to establish a regional $p$CO$_2$ climatology of the Baltic Sea, largely based on ICOS-DE SOOP *Finnmaid* surface $p$CO$_2$ observations between Lübeck-Travemünde (Germany) and Helsinki (Finland). The climatology can serve as improved input for atmosphere-ocean CO$_2$ flux estimation in this coastal environment.

## 1 Introduction

The ocean plays a major role in the global carbon cycle, and has a controlling function on the atmospheric CO$_2$ content on longer time scales (DeVries, 2022). Since the rise of atmospheric CO$_2$ concentrations during the Anthropocene, the ocean has taken up ∼25 % of the CO$_2$ released from human activities (Friedlingstein et al., 2022), with the annual uptake mainly related to the increase in the air-sea $p$CO$_2$ imbalance. The role of coastal and continental shelf waters is more complex. Apart from atmospheric CO$_2$ levels, changes in nutrient loads and organic matter supply from land, changes in weathering in the drainage basin, and even changes on the functioning and composition of biological key players on various levels can lead to changes in the inorganic carbon system and thus, the source-sink function of coastal seas (e.g., Laruelle et al., 2018; Müller et al., 2016; Carstensen and Duarte, 2019; Kuliński et al., 2022). Moreover, coastal seas provide an important conduit of land-derived carbon into the open ocean's interior (e.g., Thomas et al., 2004).

For the Baltic Sea, several attempts have been made to quantify the net CO$_2$ air-sea balance in form of a $p$CO$_2$ climatology, as well as to derive trends in surface water $p$CO$_2$, with partly contradicting results (e.g., Omstedt et al., 2009; Parard et al., 2016, 2017; Becker et al., 2021; Neumann et al., 2022; Wesslander et al., 2010). Most of the approaches either used the



output from biogeochemical models, or tried to create $pCO_2$ fields from mapped proxy data and observational data using smart extrapolation approaches. Seasonal mapping of $pCO_2$ is particularly challenging for the Baltic Sea due to its high regional and temporal variability, a salinity gradient affecting e.g. $CO_2$-equilibra, and a large seasonal amplitude caused by high net productivity in spring summer and entrainment of waters enriched in remineralization products due to mixed layer deepening in fall and winter (Schneider and Müller, 2018).

Climatologies of $pCO_2$ on a global or ocean-wide scale are an important tool for the quantification of the oceanic $CO_2$ sink in the framework of e.g. the Global Carbon Budget (Landschützer et al., 2013; Friedlingstein et al., 2022). Tailored regional analysis can help to gain insight into changes in the source-sink behaviour of distinct regions, as recently demonstrated for the northern European shelf, including the Baltic Sea (Becker et al., 2021).

A robust climatology and trend for the Baltic Sea has, apart from refining the estimate of the net $CO_2$ flux in relation to the Global Carbon Budget, several implications. As the entire Baltic Sea area belongs to the territorial waters or exclusive economic zones (EEZ) of one of the pan-Baltic nations, the air-sea $CO_2$ fluxes might be of importance for current and future carbon management and accounting schemes in the framework of national emission reduction targets (Luisetti et al., 2020). A $pCO_2$ climatology could also serve as a baseline for potential negative emissions applications, including blue carbon or coastal alkalinity enhancement (GESAMP, 2019). Knowledge of monthly $pCO_2$ fields and their variability might also help to identify and quantify the impact of perturbations and extreme events, e.g. heat waves (Humborg et al., 2019).

In this work, we build a foundation for such applications. We first present a novel extrapolation approach, followed by construction of a Baltic Sea $pCO_2$ climatology. The extrapolation approach is then evaluated and put into context with existing mapping methods. Notable features of the seasonal $pCO_2$ climatology are discussed and special attention given to the regional long-term $pCO_2$ trend before we conclude our work.

## 2 Methods

### 2.1 Extrapolation approach

For mapping from scarce observational data to spatially-filled maps of the Baltic Sea, we use an ensemble of truncated EOF reconstructions. For a more detailed description than the brief summary below, please consult appendix A.

Empirical orthogonal function (EOF) decomposition or singular value decomposition (SVD) have been used widely in atmospheric and ocean science (e.g., Lorenz, 1956; Weare et al., 1976; Weare and Newell, 1977; Hannachi et al., 2007). They can be used to efficiently reduce dimensionality of the original dataset (Lorenz, 1956; Davis, 1976; Preisendorfer, 1988; Monahan et al., 2009; Jolliffe and Cadima, 2016). Here, we use the EOF decomposition of a model dataset $\mathbf{X}$ of $pCO_2$ in the Baltic Sea to obtain spatial EOF patterns $e_i$ of $pCO_2$ variability.

In a second step, observational $pCO_2$ data $\mathbf{Y}$ are used in conjunction with these spatial patterns of variability in a truncated EOF reconstruction to constrain the EOF spatial patterns' amplitudes (Kaplan et al., 2000; Preisendorfer, 1988) and thus, to extrapolate from scarce observational data to the full domain (section A3). This step represents an optimization of a cost function $Q$ (Eq. A19), which in our case takes also the observational data uncertainty into account to avoid overfitting (section A4).





For a given EOF reconstruction with $l$ spatial modes, we thus obtain both an extrapolated field as well as an estimate of the
mapping uncertainty (section A4).

However, the choice of how many EOF modes to use for a given truncated EOF reconstruction, or at which level to truncate,
is an arbitrary choice, often inspired by a certain threshold of total variance explained (e.g., Kaplan et al., 2000).

In a third step, we therefore use an ensemble approach to circumvent this problem: Instead of a single EOF reconstruction
with a fixed number of modes $l$, we use a series of EOF reconstructions from just one mode up to a maximum number of
$l_{max}$ modes (section A5). This series is then combined by a weighted ensemble at each location with the individual EOF
reconstruction's mapping uncertainties as weights (Eqs. A25, A26; section A6).

Note that the weights are spatially resolved and depend on the available data constraints, i.e., the ensemble weights provide
for locality, which includes adaptation of the mapping's spatial scales to the data constraint density, thus providing for a more
robust extrapolation and more realistic uncertainty estimates than with a fixed number $l$ of EOF modes.

We thus obtain an ensemble mean $pCO_2$ value $x_{reconstr}$, the average number of modes $\bar{l}$ used in the ensemble of reconstruc-
tions at a given location, and an uncertainty estimate $\sigma_{reconstr}$ (Eq. A28), which is the sum of ensemble-averaged mapping
uncertainty (from each truncated EOF reconstruction with $l$ from $1...l_{max}$), ensemble-averaged representational error made
by the truncation itself (from each EOF reconstruction), and the uncertainty of the weighted ensemble itself (section A6). All
these quantities are obtained without gaps on the full spatial domain of the original model dataset $\mathbf{X}$.

Finally, due to high temporal dynamics of our Baltic Sea environment, we use an expansion of the EOF reconstruction
approach to not only reconstruct the data value, but both data value and a (short-term) linear trend in order to temporally
collate (temporally extended) observations into a time-coherent, synoptic picture (section A7; Elken et al., 2019).

## 2.2 Baltic Sea $pCO_2$ climatology

Given limitations of modelled data, we aimed to produce an observation-based $pCO_2$ climatology. For this, we used (a) the
above extrapolation approach with spatial patterns based on ecosystem model data as well as (b) observations of surface $pCO_2$
from SOCAT to produce a monthly climatology of $pCO_2$ as well as of the linear, short-term $pCO_2$ trend to temporally collate
$pCO_2$ observations (section A7) within a month (see Fig. 1 for a visualization of the approach).

### 2.2.1 Spatial patterns of variability

Spatial patterns of variability $e_i$ for our extrapolation approach are based on model data from a Baltic Sea setup of the Eco-
logical ReGional Ocean Model (ERGOM version 1.2). This version of ERGOM includes a simple carbon cycle as described
in Kuznetsov and Neumann (2013) with amendments to allow for non-Redfield stoichiometry as outlined in Neumann et al.
(2022). In principle, any carbon-containing biogeochemical ocean model of choice can be used as basis for the patterns of
variability $e_i$. Nonetheless, a better carbon representation in the model likely provides better-suited $e_i$ patterns of variability.

The model data variability characteristics are illustrated in the appendix (Fig. A1). Less than 1 % of locations show a temporal
decorrelation scale $\leq 7$ days, so that we chose a weekly aggregation of the model data for variability pattern extraction. For the
climatology itself, we chose a more typical monthly resolution.



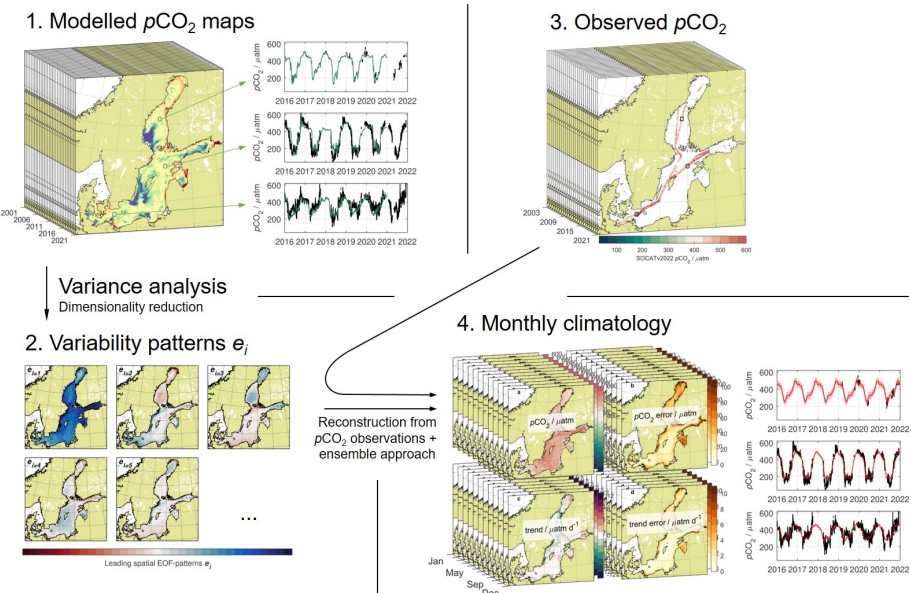

**Figure 1.** Approach to build a regional $pCO_2$ climatology of the Baltic Sea. (1) $pCO_2$ data from an ecosystem model are mathematically analyzed for dominant patterns of variability $e_i$ in a truncated EOF analysis. (2) Patterns of variability and (3) $pCO_2$ observations of a given month are used to reconstruct Baltic Sea $pCO_2$ distributions $\overline{\mathcal{X}}$, which are combined through an ensemble approach into (4) a monthly climatology of surface $pCO_2$ incl. $pCO_2$ error estimate and (short-term) $pCO_2$ trend incl. $pCO_2$ trend error estimate. For illustration, time series are given for three sample locations for model $pCO_2$ (green), $pCO_2$ observations (black), and climatological $pCO_2$ (red).

ERGOM has been shown to adequately mirror observations of the large scale nitrate, phosphate, oxygen, and carbon distribution (Neumann et al., 2022; Eilola et al., 2011; Neumann et al., 2015). However, shortcomings still exist in the exact magnitude and timing of phytoplankton carbon uptake and release throughout the seasons (Neumann et al., 2022).

From an ERGOM version 1.2 model run from 1948 – 2020 we used the last 20 years of modelled surface $pCO_2$ from 2001 to 2020 averaged into 1044 weekly means. The model run has a horizontal resolution of 3 nautical miles, which yields a dataset $\mathbf{X}$ with 12010 grid points $m$ for the Baltic Sea area South of the Skaggerak (i.e., HELCOM subbasins 2 to 17, HELCOM Secretariat, 2017) and 1044 time steps $n$.

From these model data $\mathbf{X}$, patterns of variability $e_i$ were extracted by empirical orthogonal function (EOF) analysis (see 100   previous section for details). The DINEOF analysis of our model data retained 224 EOF modes with a minimal cross-validation error of 22.6 $\mu$atm to the model data and a total explained variance of 98.6 %. The spatial patterns $e_i$ have the same 3 n.m. resolution as the model data.

### 2.2.2    Surface $pCO_2$ Observations

Surface $pCO_2$ measurements in the Baltic Sea were obtained from SOCAT version 2022 (Pfeil et al., 2013; Bakker et al., 105   2016, https://www.socat.info), which collects surface $pCO_2$ data from underway observations of ships of opportunity (SOOPs)





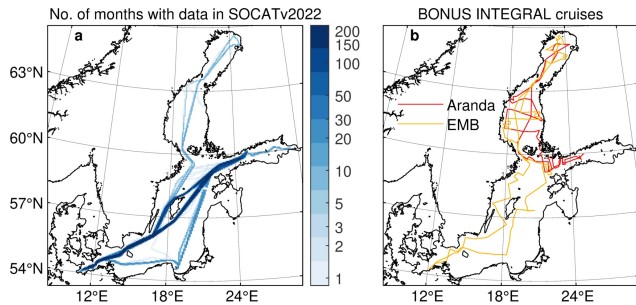

**Figure 2.** (a) Map of the Baltic Sea with number of individual months with $pCO_2$ observations available in SOCATv2022 between June 2003 and Dec. 2021 (max. 223 months). (b) Cruise tracks of BONUS INTEGRAL cruises on RV *Aranda* in Feb./Mar. 2019 and RV *Elisabeth Mann Borgese* in May/June 2019.

or research vessels. All observations are based on $CO_2$ measurements in air equilibrated with sea surface waters (Körtzinger et al., 1996; Pierrot et al., 2009) with a typical accuracy of $2-5$ $\mu$atm, in some cases $\leq 10$ $\mu$atm, which is indicated by the respective quality flag A – E (Pierrot et al., 2009; Bakker et al., 2016).

Here, Baltic Sea $pCO_2$ data for the period June 2003 to Dec. 2021 were used. They were either from the ICOS-DE SOOP
*Finnpartner/Finnmaid* line (Schneider et al., 2006; Gülzow et al., 2011) between Lübeck-Travemünde and Helsinki, which covers the Southern and Central Baltic Sea, or the ICOS-SE SOOP *Tavastland* line between Lübeck-Travemünde and Oulu/ Kemi, which additionally covers the Northern basins starting from 2019 (Fig. 2a). Additional surface $pCO_2$ data originated from RV *Aranda* cruise ARA04/2019 in Feb./Mar. 2019 and RV *Elisabeth Mann Borgese* cruise EMB214 (Rehder et al., 2021) in May/June 2019 (Fig. 2b), which were performed as part of the BONUS INTEGRAL project. Data processing and quality
control followed the SOCAT guidelines. In total, data coverage in the Southern and Central Baltic Sea is high, with locally up to 189 out of 223 months covered (i.e., June 2003 to Dec. 2021). Data in the Northern Baltic Sea, however, is only available since Feb. 2019 with observations during a total of 15 out of 35 months.

### 2.2.3 Monthly climatology construction

For every month $t$ of the 189 months with observations, the observations were centred temporally on the 15th of each month
as $t_\circ$ (see section A7) so that the extrapolation approach provides a field $\overline{\mathcal{X}}$ with the spatial distribution of

- the reconstructed $pCO_2$ ($x_{t,\mathrm{reconstr}}$, Eq. A27) at $t_\circ$,

- the $pCO_2$ error estimate ($\sigma_{t,\mathrm{reconstr}}$, Eq. A28) at $t_\circ$,

- the (short-term) $pCO_2$ trend at $t_\circ$,

- an error estimate on the $pCO_2$ trend at $t_\circ$, and

- the average number of patterns $e_i$ used in the reconstruction ($\bar{l}$, Eq. A25)



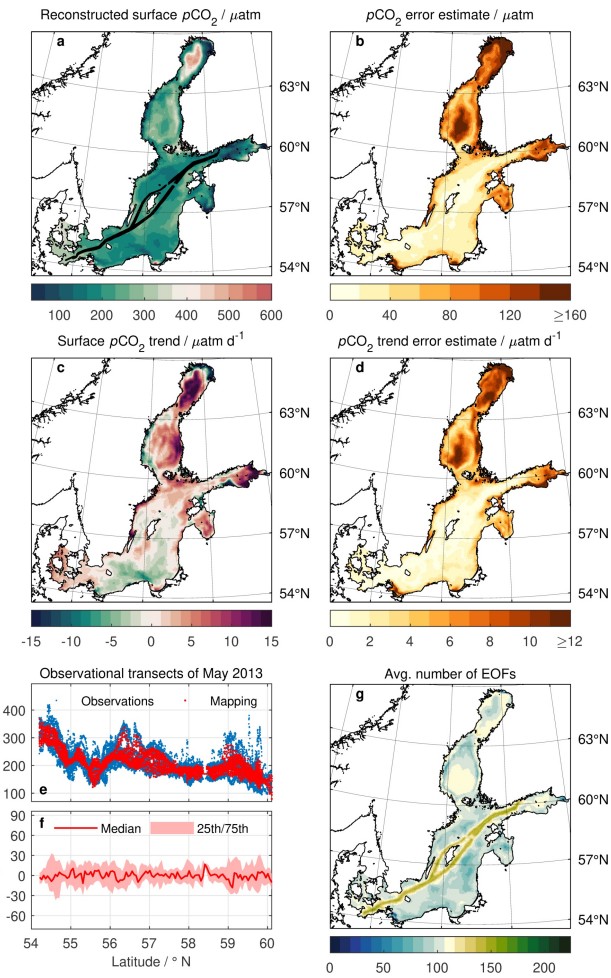

**Figure 3.** Monthly fields $\overline{\mathcal{X}}$ of surface $pCO_2$ reconstruction for May 2013: (a) Reconstructed $pCO_2$ $x_{t,\mathrm{reconstr}}$ with observation locations given as black dots. (b) $pCO_2$ error estimate $\sigma_{t,\mathrm{reconstr}}$ with small values nearby observations. (c) $pCO_2$ trend and (d) its error estimate. (e) Observed (blue) and reconstructed $pCO_2$ (red) against latitude as well as (f) their difference as median (solid line) and interquartile range (pale shading). (g) Average number of patterns $\bar{l}$ of the reconstruction: Above average $\bar{l}$ indicates use of high-order, smaller scale patterns $e_i$, notably nearby data constraints from observations, whereas below average $\bar{l}$ indicates areas where low-order, larger scale patterns $e_i$ dominate in the reconstruction ensemble (see Eq. A26). Compare Fig. A2 for a reconstruction without temporal trend (section A7).

.

for each of month with observations (e.g., Fig. 3 for $\overline{\mathcal{X}}$'s of May 2013 as illustration).

As not all months between Jun. 2003 and Dec. 2021 are covered and to provide a representative picture of seasonality, we combined the thus obtained 189 monthly maps $\overline{\mathcal{X}}$ into a mean monthly climatology $\mathbf{Y}$ with size $m \times 12$.



For a given location $m$, $\chi_m$ denotes the (reconstructed) time series of $\overline{\mathcal{X}}$ (with size $1 \times 189$) and $y_m$ the time series of the
mean monthly climatology $\mathbf{Y}$ (with size $1 \times 12$), respectively. The monthly means $y_m$ were then obtained by:

$$w_m \cdot \chi = w_t \cdot \left( \mathbf{O}_m^{\mathbf{T}} y_m^{\mathbf{T}} + g_m \cdot (t - t_{\text{ref}}) \right) \text{ with} \tag{1}$$

$$w_m = 1/\sigma^2_{m,\text{reconstr}} \;, \tag{2}$$

where $w_m$ is the time series of inverse variance weights (Eq. 2) with size $1 \times 189$ at the given location $m$, $\mathbf{O}_m$ a time series
operator or matrix of size $12 \times 189$ that selects the monthly mean $y_m$ corresponding to the month of $\chi_m$, $t - t_{\text{ref}}$ the time
difference (in decimal years) between each of the 189 monthly $\overline{\mathcal{X}}$ and a reference time, $t_{\text{ref}}$, and $g_m$ an extra degree of freedom
to allow for a linear long-term trend in $\overline{\mathcal{X}}$ for each location $m$ (e.g., surface $p\mathrm{CO}_2$). As reference time, the middle of 2013 was
chosen, which closely corresponds to the mean date of all observations.

Eq. 1 represents a system of 189 linear equations with 13 degrees of freedom to calculate 12 weighted mean values $y_m$ and a
long-term trend $g_m$. Through the inverse variance weights (Eq. 2), monthly maps $\chi_m$ with better constrained data, i.e., smaller
variance of reconstruction $\sigma^2_{m,\text{reconstr}}$ at the given location $m$ (and for the given month), obtain preference in the weighted
mean (Eq. 1).

When done for each point $m$, one obtains the mean seasonal climatology $\mathbf{Y}$ (with size $m \times 12$), centered on 2013, as well
as a map of its mean linear (long-term) trend $\mathbf{G}$ (with size $m \times 1$) for every of the five fields $\overline{\mathcal{X}}$ discussed.

## 3 Results

### 3.1 Extrapolation approach

The mapping approach gives fully filled fields on the entire spatial domain from scattered observational data. The mapped
$p\mathrm{CO}_2$ is in a similar range and frequency distribution as observations (Fig. 4a). The $p\mathrm{CO}_2$ error estimate is reduced in spatial
vicinity to observations. In contrast, $p\mathrm{CO}_2$ error is markedly increased both near shore as well as in subbasins not covered by
observations, e.g. the Northern basins or the Gulf of Riga for a SOOP *Finnmaid*-based reconstruction (e.g., Fig. 3b). Still, $p\mathrm{CO}_2$
values remain within reasonable margins even in those areas (e.g., Fig. 3a). We can not observe a tendency of the mapping
approach to give extreme values or outliers in absence of observations (compare Fig. 4a). The mapped fields are smooth and
without spatial gaps or discontinuities.

Addition of a linear temporal trend to collate observations to a common time $t_\circ$ helps to reduce the mismatch between
temporally-spread observations and a full-domain mapping of a dynamic coastal system (compare, e.g., Fig. 3e, f and Fig. A2c,
d, respectively).

However, for about half the grid points of the 189 monthly mappings, the magnitude of the short-term temporal $p\mathrm{CO}_2$ trend
(e.g., Fig. 3c) at a given location $m$ is insignificant, i.e., within its trend error estimate (e.g., Fig. 3d). At the same time, a
considerable portion of the significant $p\mathrm{CO}_2$ trends are found at grid points $m$ in the direct footprint of observations where
the trend error estimates are small in the first place. This demonstrates that a monthly temporal resolution is sufficient for
construction of a Baltic Sea $p\mathrm{CO}_2$ climatology with our approach.





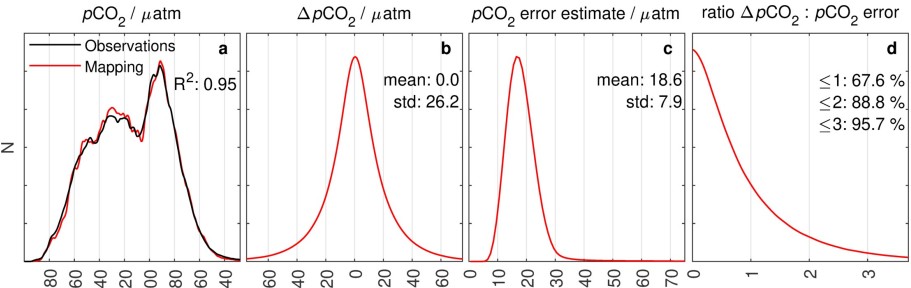

**Figure 4.** Histograms for: (a) $pCO_2$ observations (black) as described in section 2.2.2 and $pCO_2$ reconstructed at the observation times and locations from the 189 monthly mappings (red). Data are highly correlated ($R^2$) and they show a similar distribution. (b) The $pCO_2$ difference between reconstructed and observed $pCO_2$. There is no bias in the reconstruction. (c) The $pCO_2$ error estimate $\sigma_{\mathrm{reconstr}}$ of the reconstruction at the observation times and locations. (d) The ratio between $pCO_2$ difference and $pCO_2$ error estimate, where a ratio $\leq 1$ means that the observed $pCO_2$ is within $1 \times \sigma_{\mathrm{reconstr}}$ of the mapped $pCO_2$.

## 3.2 Quality of interpolation and extrapolation

To assess the quality of obtained fields, we consider (a) the mapped result against the original observations, and (b) a comparison of mappings from concurrent subsets of observations.

For the first aspect, we consider the residual $pCO_2$ against SOCAT observations for all 189 monthly mapped fields from the climatology construction (e.g., Fig. 3e, f). We use typical metrics for interpolation methods (correlation, bias, standard deviation). As the monthly fields $\overline{\mathcal{X}}$ were constructed from those observations, this assesses primarily to which degree the data constraints were taken into account for the mapping, e.g., the balance between point-to-point reproduction and smoothing with the given patterns of variability $e_i$.

The comparison shows highly correlated $pCO_2$ data ($R^2 = 0.95$), a similar $pCO_2$ distribution (Fig. 4a), and an unbiased
mapping with a standard deviation of 26 $\mu$atm (Fig. 4b). This, includes both misfits of the mapped fields as well as variations within observations of a given grid point location (compare spread between observations and mapping in Fig. 3e or Fig. 5c, f). The error estimate is of a similar magnitude (Fig. 4c) and well co-located, i.e., residuals are within $1 \times$ the error estimate in 68 % of cases (Fig. 4d).

For the second aspect, we reconstruct the surface $pCO_2$ distribution independently from SOOP *Tavastland* and SOOP *Finn-*
*maid* data, respectively, for the same time period. We note, however, that the selection of "independent" data is to some degree arbitrary (e.g., portions of a given SOOP transect could be seen as "independent", or data from SOOP vessels vs. research vessels, or data from one week to data from the next week, etc.), with impact on thus derived statistics. We therefore stepped back from a quantitative attempt but provide just a qualitative picture (Fig. 5).

Evaluated on the original data (i.e., SOOP *Tavastland* reconstruction on SOOP *Tavastland* observations and SOOP *Finnmaid*
reconstruction on SOOP *Finnmaid* observations, respectively), the above statistics against original observations are confirmed:





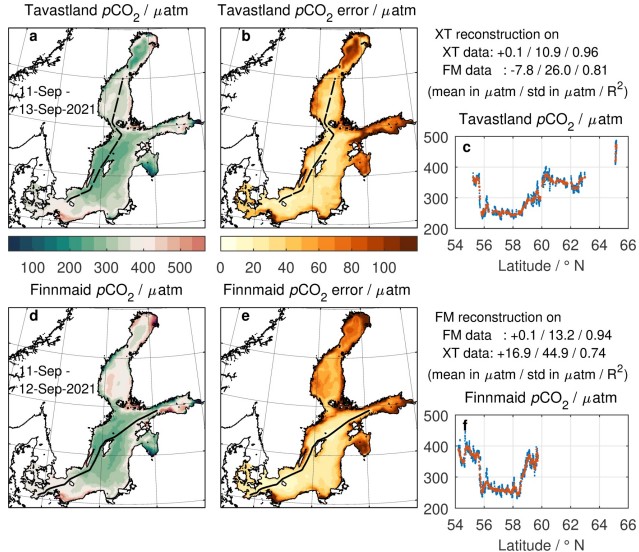

**Figure 5.** Surface $pCO_2$ reconstruction for 11 – 13 Sept. 2021 based on concurrent SOOP *Tavastland* (top) and SOOP *Finnmaid* transects (bottom), respectively. (a,d) Surface $pCO_2$ distribution and (b,e) estimated error are shown with locations of observations denoted as black dots. (c,f) Observed (blue) and reconstructed $pCO_2$ (red) against latitude as well as evaluation metrics (above each panel) for both reconstructed $pCO_2$ fields on both datasets, i.e., SOOP *Tavastland* (XT) and SOOP *Finnmaid* (FM) reconstruction evaluated at the observed locations of the XT and FM, respectively.

Observations are reproduced unbiased and with some degree of spatial smoothing, i.e., with some small-scale scatter but without large-scale residual pattern (Fig. 5c, f).

Comparison of both mappings in areas of extrapolation, i.e., away from observations used to produce the $pCO_2$ fields, shows a similar $pCO_2$ distribution in the Western and Central Baltic Sea (Fig. 5a, d). The estimated error is on the order of 15 $\mu$atm in the vicinity of observations and increases with distance to around 20 – 30 $\mu$atm in the open Baltic and to higher values nearshore (Fig. 5b, e). Differences exist in the $pCO_2$ field both in the Northern basins as well as in the Gulf of Finland, where either SOOP *Finnmaid* or SOOP *Tavastland* provide no observational constraints to the mapping approach (Fig. 5a, d). In both cases, this is mirrored by elevated $pCO_2$ error estimates (Fig. 5b, e), i.e., areas of extrapolation are properly assigned a higher $\sigma_{\mathrm{reconstr}}$ by the mapping approach.

## 3.3 Baltic Sea $pCO_2$ climatology

### 3.3.1 $pCO_2$ distribution

The mean seasonal $pCO_2$ distribution $x_{\mathrm{reconstr}}$ matches previously published seasonal cycles of surface $pCO_2$ in the Western and central Baltic Sea (Schneider and Müller, 2018): A strong spring bloom $pCO_2$ drawdown occurs between March and May to a low around 200 $\mu$atm in May, some slight increase in June and a second summer bloom low in July, followed by relaxation



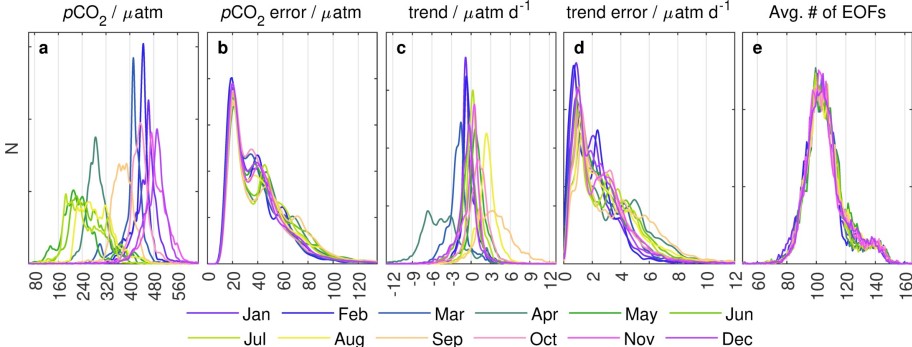

**Figure 6.** Smoothed histograms for the five output fields of the climatology $\mathbf{Y}$: (a) reconstructed $pCO_2$ $x_{t,\mathrm{reconstr}}$, (b) $pCO_2$ error estimate $\sigma_{t,\mathrm{reconstr}}$, (c) $pCO_2$ short-term trend, (d) error estimate on the $pCO_2$ short-term trend, and (e) average number of patterns $\bar{l}$ used in the reconstruction. The colour code indicates the respective month. $pCO_2$ follows a typical seasonal cycle, while the $pCO_2$ error estimate and number of patterns show no apparent seasonal variations. $pCO_2$ trends show higher magnitudes during spring (Mar./Apr.) and late summer (Aug./Sep.), which is mirrored by enhanced $pCO_2$ trend error estimates in these months.

towards atmospheric levels in September/October and supersaturated levels during late autumn and winter, peaking around 500 $\mu$atm in December (Fig. 6a).

Regionally, the Western basins lead the mean seasonal cycle, while the central Gotland basin and the Gulf of Finland trail behind. The productive season is even shorter in the Northern basins, with the major $pCO_2$ drawdown in the Bothnian Sea occurring in June (Fig. A3 and A4, 1st column). Similarly, the seasonal amplitude is less pronounced in the Western or

Northern basins than in the Gotland basin, and is most intense in the Gulf of Finland. For the Northern basins, this is the first fully seasonal climatology from $pCO_2$ observations.

The $pCO_2$ error estimate $\sigma_{\mathrm{reconstr}}$ shows a minimum of 12 $\mu$atm. Close to observations, the $pCO_2$ error estimate of the mean seasonal climatology is on the order of 20 $\mu$atm and as such close to the value of the individual mappings (compare Fig. 6b vs. Fig. 4c). Further away, $\sigma_{\mathrm{reconstr}}$ increases up to a level around 90 $\mu$atm (95th percentile), with a spatial preference for high

$\sigma_{\mathrm{reconstr}}$ in the Northern basins as well as in the Gulf of Riga, Oder bight, and other sheltered, near-shore areas (Fig. A3 and A4, 2nd column).

### 3.3.2  $pCO_2$ short-term trends

In general, the short-term $pCO_2$ trends follow the mean seasonal $pCO_2$ cycle with on average negative trends, i.e., decreasing $pCO_2$, during winter and spring. Positive trends prevail during late summer and autumn (Fig. 6c) with strongest trend

magnitudes occurring in the Northern Baltic Proper and the Gulf of Finland (Fig. A3 and A4, 3rd column).

However in 8 out of 12 months, the majority of estimated trends are insignificant, i.e., more than 70 % are smaller than their error estimate, indicating that inclusion of a short-term trend in the mapping may not be required for these months. Only in March and April as well as in August and September, the majority of $pCO_2$ trends is significant (despite at the same time





elevated $pCO_2$ trend error estimate in some areas, see Fig. 6d). I.e., significant short-term $pCO_2$ trend magnitudes fall together
with the strong springtime $pCO_2$ drawdown as well as with autumnal mixed layer deepening and entrainment of high-$pCO_2$
waters (Fig. 6; more in Jacobs et al., 2021).

Like for the $pCO_2$ distribution, trend error estimates are increased in the Northern basins as well as for the Gulf of Riga,
Oder bight, and other sheltered, near-shore areas (Fig. A3 and A4, 4th column).

### 3.3.3   Number of patterns or mapping scales

The average number of patterns $\bar{l}$ per month can be seen as a qualitative indicator to show preference for reconstructions with
larger or smaller spatial scales, respectively, in the weighted mean (Eq. A25) of the reconstruction.

For the climatology $\mathbf{Y}$, $\bar{l}$ shows a maximum around 105 patterns (Fig. 6e). This level is at about 45 % of the maximum
number of patterns, i.e., indicating only a slight preference for reconstructions with larger than average scales in the weighted
mean. However, a notable fraction with higher number of patterns $\bar{l}$ with peak around 140 patterns exist (Fig. 6e). These are
located in vicinity to observations, where preference in the weighted mean is on reconstructions with stronger small scale
features (see, e.g., Fig. 3g).

There is no seasonal imprint to the average number of patterns $\bar{l}$, which is in contrast to the $pCO_2$ as well as the $pCO_2$
short-term trend distribution (Fig. 6).

### 3.4   Long-term $pCO_2$ trend

The mean climatology $\mathbf{Y}$ is constructed from the 189 monthly mappings $\overline{\mathcal{X}}$ and a linear long-term trend $\mathbf{G}$ (see Eq. 1 and
section 2.2.3). For $pCO_2$, the analysis shows a significant long-term $pCO_2$ increase of +1.4±0.5 $\mu$atm yr$^{-1}$ (1 $\sigma$) in the
Southern Baltic Sea from the Great Belt to the Bornholm basin and Southern parts of the Gotland basin (Fig. 7). A similar
increase (+1.5±0.7 $\mu$atm yr$^{-1}$) is obtained in selected regions of the Northern Baltic Proper, the Åland Sea and the Gulf of
Finland. A positive trend in $pCO_2$ is also visible in coastal areas along the Finnish coast in the Northern basins as well as
in the Gulf of Riga. However, multi-year observations in those areas are unavailable to support a long-term $pCO_2$ trend (see
section 2.2.2; Fig. 2). In all other areas, the data available do not indicate a significant long-term trend, i.e., one that is outside
the 68.3 % (1 $\sigma$) confidence bound. This includes in particular the Western and Eastern Gotland basin, but also the Northern
basins (Fig. 7), where the observational time series is likely too short to allow for the detection of a significant long-term trend
(Fig. 2).

## 4   Discussion

### 4.1   Extrapolation approach

The extrapolation approach uses an EOF analysis of the data covariance $\mathbf{Q}$ as foundation. When working with an EOF recon-
struction, one needs to bear in mind the purely mathematical nature of the EOF modes' $e_i$ construction: They are built by (1)


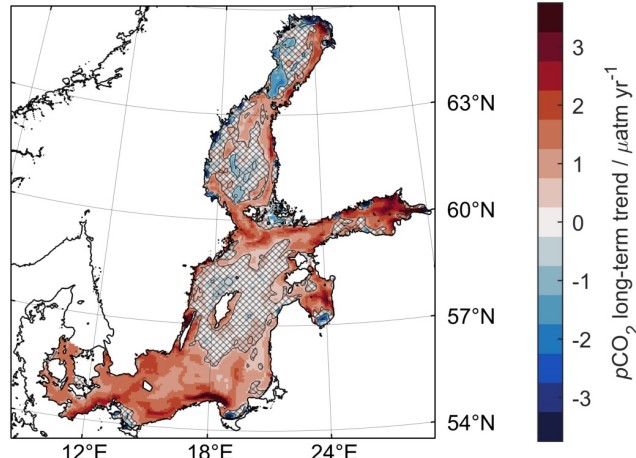

**Figure 7.** Long-term trend **G** of surface $p$CO$_2$ from the monthly mappings and the climatology construction (Eq. 1). Areas where the long-term trend is not significant are cross-hatched. In the Southern Baltic Sea, a significant increase of +1.4 $\mu$atm yr$^{-1}$ is obtained from the observations. Selected parts of the Northern Baltic Proper, the Åland Sea and the Gulf of Finland give a similar increase of +1.5 $\mu$atm yr$^{-1}$, while the Western and Eastern Gotland basin show no significant long-term trend.

maximizing the (partial) variance they explain and by (2) being orthogonal to all previous EOF modes (Preisendorfer, 1988;

Monahan et al., 2009; Dommenget, 2015). They may therefore indicate some causal connection within the original data **X**, which requires confirmation by other means. They may, however, just as well group some portion of (partial) variance in one part of the domain with some portion of (partial) variance in another part of the domain just to mathematically maximize the amount of (partial) variance of the given EOF mode $e_i$. This can give rise to apparent EOF mode "teleconnections", where one part of the data or spatial domain are seemingly tightly coupled with another part of the data, e.g., for the second EOF

mode $e_{i=2}$, which has always a dipole pattern. Often, there is a temptation to attribute such couplings or "teleconnections" to a driving mechanism. However, given its origin, patterns of EOF modes should be interpreted with great caution (and only with ancillary, supporting data) and best seen of mathematical rather than of mechanistic nature (Monahan et al., 2009; Dommenget, 2015). We therefore did not try to assign physical mechanisms or drivers to specific EOF patterns $e_i$ in our work.

The use of EOF modes as basis for extrapolation ensures that the extrapolated map covers the full spatial domain, that it is

continuous/gap-free, and that it is discontinuity-free.

A key aspect of truncated EOF reconstructions is the number $l$ of significant modes included. Our cross-validation approach provided a relatively large number of $l_{\max} = 224$ modes. The relatively large number of modes is a prerequisite for a small representational error **P′**, i.e., for the covariance not resolved by the collection of truncated EOF modes, which gives a lower bound on the uncertainty of the mapping $\sigma_{\mathrm{reconstr}}$ (Eq. A28). In our case, the average $\sigma_{\mathbf{P′}}$ when using all 224 patterns $e_i$

amounts to 14.7 $\mu$atm. If only the first 50 patterns $e_i$ were included, the average representational error would be as high as 29.1 $\mu$atm.

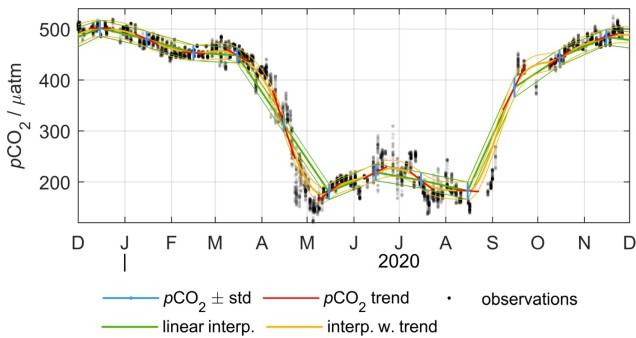

**Figure 8.** Difference between monthly point-by-point interpolation (green) or interpolation with a trend (orange) in a dynamic coastal system. Example data are from the Northern Baltic Proper (ca. 021° 00' E, 58° 45' N) with $pCO_2$ observations **X** shown in black. They are the basis for monthly mapped fields $\overline{\mathcal{X}}$ of surface $pCO_2$ (blue) as well as for a linear (short-term) $pCO_2$ trend (red). Compared to a linear interpolation between monthly $pCO_2$ values (green), interpolation including the trend information (orange) gives a better reproduction of the observed dynamics particularly in spring and autumn. A cubic Hermite spline was used for the calculation (Appendix B).

A reconstruction with a small number of modes provides for a more uniform, large-scale homogeneous field, where gaps in the data are filled by the large-scale picture. However, such reconstructions may lack the flexibility to reproduce real features of the observations, e.g., through too strong smoothing. Conversely, a large number of modes provides for a fine, small-scale field with high flexibility. However, features in some areas without nearby observations may be badly constrained with the risk of "ghost" signals.

By using an ensemble of reconstructions that cover the entire range of $1 \dots l_{\mathbf{max}}$ modes and by using the mapping variance $\sigma_{\mathbf{P}}^2$ as weights (Eqs. A25, A26), we find an optimal trade-off in our mapping approach, where both aspects are balanced. This way, the scales used in the mapping are adjusted spatially according to the distribution of observations as constraints.

The cost function of the mapping approach minimizes the residual between observations and mapped data (Eq. A19). This way, a synoptic $pCO_2$ mapping is obtained. If observations, however, are temporally extended compared to the system's temporal dynamics, the $pCO_2$ mapping can be distorted and not represent a synoptic picture, depending on the spatial and temporal pattern by which observations are obtained (Elken et al., 2019).

The introduction of a temporal trend at each location (section A7) as additional degree of freedom reduces such distortion caused by sampling. It both improves the fit to the data (compare Fig. A2c, d vs. Fig. 3e, f) and provides extra information:

– For a given mapping, a strong or weak trend indicates where temporal dynamics are high or low and informs on where frequency of observations should be enhanced or not, respectively.

– For a time series of mappings, information on both the value and trend allows for a more accurate interpolation by cubic Hermite spline (Appendix B) compared to a standard, linear interpolation of the values alone (e.g., Fig. 8).





For our mapping of surface $p\mathrm{CO}_2$ in monthly intervals, we conclude that the inclusion of a temporal trend in the mapping is required, particularly for spring and autumn. For months without strong trends, we do not see adverse effects in the mapped $p\mathrm{CO}_2$ compared to an extrapolation without inclusion of a temporal trend.

The dynamics of the $p\mathrm{CO}_2$ field are prescribed by the model data covariance matrix $\mathbf{Q}$ (Fig. A1a). The closer the model and real-world variability match, the easier it is for the mapping to reflect reality and to provide a most realistic picture. Nonetheless,

also for a model dataset $\mathbf{X}$ that may lack in some real-world features or their magnitude, the observational constraints provide for an improvement in the mapped field compared to the original model.

The mapping error estimates are elevated where observations are scarce or dynamics are high to start with. For our $p\mathrm{CO}_2$ mapping, the model covariance $\mathbf{Q}$ is the origin for a pronounced coast-basin difference in the $p\mathrm{CO}_2$ variance caused both by physical and biological drivers in the model. This prescribed data covariance distribution is retained in the mapping error

estimates $\sigma_{\mathrm{reconstr}}$. E.g., they are enlarged in dynamic river plumes, in the sheltered, shallow Gulf of Riga, or in other near-shore areas, unless there are observations to constrain dynamics and thus reduce $\sigma_{\mathrm{reconstr}}$. In regions with few observations, $\sigma_{\mathrm{reconstr}}$ approaches the prescribed data variance from $\mathbf{Q}$ (Fig. A1a).

We do not observe a seasonality in the number of patterns $\bar{l}$ of the mapping (Fig. 6e). At the same time, the $p\mathrm{CO}_2$ value varies strongly between seasons. In addition, there is a slight preference for observations to occur in spring, summer, and autumn over

winter months. However, there is no discernible difference in the spatial coverage of $p\mathrm{CO}_2$ observations between seasons (data not shown). The aseasonality of $\bar{l}$ can be explained by the "large-scale" mapping error $\sigma_{\mathbf{P}}$, used for the ensemble weights (Eq. A26), to only depend on the distribution of samples $\mathbf{H}_t$ and their respective observational error estimates $\mathbf{S}$. Therefore, the extrapolation approach does not discriminate or distinguish by ($p\mathrm{CO}_2$) value and there is no seasonal imprint in the number of patterns.

## 4.2    Comparison with other $p\mathrm{CO}_2$ mapping approaches

Observation-driven extrapolation approaches can be grouped into different categories: (a) one, where sparse data points are extrapolated based on some statistic metric of influence, e.g., a spatial decorrelation length or time scale, and (b) one where parametrizations with proxy data in combination with filled fields of those proxies are used for extrapolation (Rödenbeck et al., 2015).

Our mapping approach belongs to the first category. For a given month or time window, we use the available $p\mathrm{CO}_2$ data directly and 1:1 without added transformation to inform the extrapolated $p\mathrm{CO}_2$ map. As (spatial) metric of influence, the variability patterns are used. This gives a direct link between observations of a given month or time window and the mapped field. Conversely, if there are no $p\mathrm{CO}_2$ observations in a given time window, this directly translates to a (time) gap in the mapped fields for any approach of the first category.

Most previous $p\mathrm{CO}_2$ mappings in the Baltic Sea belong to the second category (e.g., Parard et al., 2016; Becker et al., 2021). In a first step, they use all available $p\mathrm{CO}_2$ data together with co-located proxies (e.g., location, sea surface temperature, chlorophyll $a$ concentration, water depth, distance to coast, season, ...) to establish a relationship or parametrization between proxies and $p\mathrm{CO}_2$. In a second step, distributions of those proxy parameters are used to establish a $p\mathrm{CO}_2$ distribution in time





and space. Here, $p\text{CO}_2$ data of a given month do *not* directly inform the extrapolated $p\text{CO}_2$ map, but only indirectly through

the established relationship using all data. Conversely, if there are no $p\text{CO}_2$ observations in a given time window but proxy data are available, they can be used to estimate a $p\text{CO}_2$ map nonetheless, i.e., (temporal) gap filling is possible with the two-step procedure of approaches of the second category.

The nature of the relationship or parametrization used can vary, e.g.,

–   using multi-linear regressions (in sub-regions) (MLR methods: E.g., Schuster et al., 2013; Becker et al., 2021), or

–   using self-organizing maps (SOM) for regionalization together with a regression method like a feed-forward neural network (SOMFFN method: Landschützer et al., 2013) or a linear regression component (SOMLO method: Sasse et al., 2013; Parard et al., 2016).

The principle of those algorithms, however, which all belong to the second category, remains the same.

As a drawback, mapped $p\text{CO}_2$ fields of the second category rely on the validity of the proxy relationship and equally

important on good proxy input data. In some cases, e.g., satellite-based sea surface temperature or chlorophyll *a* concentration, proxies may be (seasonally) prone to artefacts, e.g., due to stronger cloud cover in winter months. Such artefacts then directly translate into mapped $p\text{CO}_2$ fields and may create unrealistic patterns or out of range values (e.g., Parard et al., 2016). A mapping approach of the first category such as the one presented here, which uses the observations directly, is not affected by external-source data quality and mapped $p\text{CO}_2$ fields are therefore less prone to show out-of-range values (Fig. 4a).

### 4.3   Baltic Sea $p\text{CO}_2$ climatology

The construction of a $p\text{CO}_2$ climatology for the entire Baltic Sea has its bottleneck in the temporal coverage of the Northern basins, i.e., the Bothnian Sea, the Quark, and the Bothnian Bay. While the central and Southern Baltic Sea has ca. 85 % of all months since June 2003 covered, observations in the Northern basins start only in Feb. 2019 with 43 % monthly coverage until the end of 2021. While the extrapolation approach provides fully filled maps of $p\text{CO}_2$, including the Northern basins, also

for months pre-Feb. 2019, they are associated with larger uncertainties, i.e., limited explanatory power in the Northern basins (e.g., Fig. 3b, Fig. 5e). To not create a wrong impression of seasonality or interannual variability, especially if the mapping uncertainty fields $\sigma_{t,\text{reconstr}}$ are not respected by a casual user, we chose to provide a mean seasonal climatology $\mathbf{Y}$. Here, the weighting of the climatology (Eq. 1) ensures the preference of reconstructions with proper observation constraints (i.e., Feb. 2019 and onwards for the Northern basins) over reconstructions without (i.e., pre-Feb. 2019).

Provision of a mean climatology $\mathbf{Y}$ has another advantage: It can serve as a seasonal baseline of regional $p\text{CO}_2$ evolution that enables assessment of interannual variations, e.g., with respect to timing or amplitude.

### 4.4   Long-term $p\text{CO}_2$ trend

The long-term trend of surface $p\text{CO}_2$ in Baltic Sea surface waters has been studied before: Wesslander et al. (2010) describe an increase in monthly $p\text{CO}_2$ (ca. +20 $\mu$atm yr$^{-1}$) for the 1993–1998 period in the Eastern Gotland basin, which seems mostly

driven by less intense summer $p\text{CO}_2$ minima (from ca. 50 $\mu$atm in 1993 to ca. 250 $\mu$atm in 1998). For the longer period





1993–2009, they do not detect a significant trend. While Wesslander et al. (2010) use $pCO_2$ calculated from pH and alkalinity, all further studies are based on the same surface $pCO_2$ observations archived in SOCAT (Bakker et al., 2016), albeit using different methods and time extents:

Laruelle et al. (2018) found an increase of +2.9±2.4 $\mu$atm yr$^{-1}$ (1 $\sigma$) for winter-time $pCO_2$ from 1995–2011 in the Southern and Central Baltic Sea. At the same time, they see a similar spatial gradient as in our work (Fig. 7) with stronger increase in the South-west and zero or even negative trend in the Northern Gotland basin.

Schneider and Müller (2018) describe an increase of +4.6 to +6.1 $\mu$atm yr$^{-1}$ (1 $\sigma$ between 0.6−1.5 $\mu$atm yr$^{-1}$) for different areas for the 2008–2015 period. However, their analysis did not deseasonalize the observations prior to linear trend analysis. During their analyzed period, there is an increase in observations during autumn and winter (with typically high $pCO_2$) and

a decrease in early spring observations (with typically low $pCO_2$), while late-spring and summer observations have a similar data coverage over the years. Their relatively high $pCO_2$ trend estimate therefore seems to be partially caused by a seasonal shift in data availability that was left unaccounted for in the analysis.

Becker et al. (2021) derived a trend of +2.05±0.12 $\mu$atm yr$^{-1}$ for the Southern Baltic Sea (South of 56° N), and of +1.84±0.21 $\mu$atm yr$^{-1}$ for the Central Baltic Sea (between 56° N and 61° N) for the period 1995–2016. However, they

observe a stronger trend in earlier years (starting in the 90's) than for later periods (starting in the mid-2000's).

Our analysis covers the more recent period 2003–2021 and gives a significant trend of +1.4±0.5 $\mu$atm yr$^{-1}$ (1 $\sigma$) in the Southern Baltic Sea (South of 56.5° N) and of +1.5±0.7 $\mu$atm yr$^{-1}$ in the Åland Sea, parts of the Northern Baltic Proper, and the Gulf of Finland (between 59° N and 61° N). The Central Baltic Sea as well the as the Northern basins themselves show no significant trend. The positive trend (+1.2±0.6 $\mu$atm yr$^{-1}$) in coastal waters of the Northern basins along the Finnish coast

as well as in the Gulf of Riga (Fig. 7) has currently no support by direct observational constraints in those areas (Fig. 2). In contrast, it likely originates from high similarity to other coastal waters (e.g., along the Southern Baltic shore) in the model data variability analysis and thus in the variability patterns $e_i$ used for mapping. As such, the $pCO_2$ increase may be realistic, but lacks field data support.

Together with the literature, our results seem to indicate a reduction in overall surface $pCO_2$ increase in the Baltic Sea during

the past decades, or even its complete absence like in the Central Baltic Sea (Fig. 7) in recent periods. While atmospheric $pCO_2$ levels are still rising (ca. +2.4 $\mu$atm yr$^{-1}$ in the past decade; Lan et al., 2022), this has implications for the $CO_2$ balance between atmosphere and Baltic Sea. It supports previous claims of coastal seas to become a less intense source of $CO_2$ or eventually to turn into (an increasing) $CO_2$ sink (Laruelle et al., 2018).

## 5   Conclusions

In this work, we developed an extrapolation approach that combines two worlds: Models, specifically the distribution and connectivity that exists in model data variability, and observations in that they provide constraints of the real world picture.

Most notable features of the approach are that it does not tend to give extreme, out-of-range values even with few data constraints and that it provides local error estimates, which reflect both underlying variability, e.g., coast-basin gradients, as





well as observational data constraints. We consider of particular merit that the extrapolation scheme adapts its spatial scales to
the amount of observations in a certain area, leading to a sound representation of less uncertainty where more data are available.

Used together with high-quality surface $pCO_2$ data from SOCAT, we established a climatology that covers the entire Baltic
Sea. Given the present data scarcity in the Northern basins of the Baltic Sea, a mean $pCO_2$ climatology is what is reliably
achievable today. With sustained and/or enhanced observations in such data poor regions, a fully monthly-resolved $pCO_2$
dataset will become realistic in the future. To this end, the recently ICOS-adopted SOOP *Tavastland* as well as SOOP *Silja
Serenade*, crossing between Stockholm and Helsinki, provide a positive outlook for uncertainty reduction.

Finally, our extrapolation approach as well as the method to establish a climatology is neither limited to $pCO_2$ nor to the
Baltic Sea. Instead, it is transferable to other areas and parameters where the present work can serve as a template.

## 6 Data availability

Surface $pCO_2$ data are available from SOCAT (https://www.socat.info, Bakker et al., 2016) and Rehder et al. (2021). ERGOM
model output data are available at https://thredds-iow.io-warnemuende.de/thredds/catalogs/projects/integral/catalog_pocNP_
V04R25_3nm_agg_time.html?dataset=IOW-THREDDS-Baltic_pocNP_V04R25_3nm_agg_time_2020-11-20-12. The clima-
tology has been submitted to Pangaea and will be made publicly available upon final publication. As interim, the submitted
data can be accessed through Bittig et al. (2023).

## Appendix A: Extrapolation approach

### A1 Notation and mathematical background

To represent a spatial dataset at a given time $t$ with $m$ spatial points, we use the data vector $x_t$ with size $m \times 1$. For a spatial
time series of $n$ times, we use the data matrix $\mathbf{X}$ with size $m \times n$, where the $n$ columns correspond to $n$ data vectors $x_t$. Data
in $x_t$ and $\mathbf{X}$ are the deviation from the space-dependent temporal mean $\overline{x}_m$. The empirical data covariance matrix $\mathbf{Q_{XX}}$ with
size $m \times m$ is given by

$$\mathbf{Q_{XX}} = \frac{1}{n-1}\mathbf{X}\mathbf{X}^{\mathrm{T}} \,. \tag{A1}$$

The singular value decomposition (SVD) or empirical orthogonal function (EOF) decomposition of the data matrix $\mathbf{X}$
decomposes $\mathbf{X}$ into a series of orthonormal factors or principal components according to

$$\mathbf{X} = \mathbf{E}\mathbf{\Sigma}\mathbf{A}^{\mathrm{T}} \,, \tag{A2}$$

where $\mathbf{E}$ is a $m \times m$ orthonormal matrix, $\mathbf{\Sigma}$ a $m \times n$ rectangular diagonal matrix, and $\mathbf{A}$ a $n \times n$ orthonormal matrix. With the
above convention on $m$ and $n$, the column vectors $e_i$ (with size $m \times 1$) in $\mathbf{E}$ represent spatial patterns while the column vectors
$a_i$ (with size $n \times 1$) in $\mathbf{A}$ represent amplitude time series. They are also called left-singular vectors and right-singular vectors,



respectively. The diagonal elements of $\boldsymbol{\Sigma}$ are the so-called singular values $s_i$ of $\mathbf{X}$. Eq. A2 can be written in vector form as

$$x_t = \mathbf{E}\boldsymbol{\Sigma}a_t = \mathbf{E}\alpha_t \ , \tag{A3}$$

where $\alpha_t = \boldsymbol{\Sigma}a_t$ is the so-called dimensional amplitude at a given time $t$.

Based on a SVD of $\mathbf{X}$ (Eq.A2), the $m \times m$ matrix $\mathbf{B} = \mathbf{X}\mathbf{X}^{\mathrm{T}}$ can be expressed as

$$\mathbf{B} = \mathbf{X}\mathbf{X}^{\mathrm{T}} = \mathbf{E}\boldsymbol{\Sigma}\mathbf{A}^{\mathrm{T}}\,\mathbf{A}\boldsymbol{\Sigma}^{\mathrm{T}}\mathbf{E}^{\mathrm{T}} = \mathbf{E}\,\boldsymbol{\Sigma}\boldsymbol{\Sigma}^{\mathrm{T}}\,\mathbf{E}^{\mathrm{T}} = \mathbf{E}\,\tilde{\boldsymbol{\Lambda}}\,\mathbf{E}^{\mathrm{T}} \ , \tag{A4}$$

where $\tilde{\boldsymbol{\Lambda}}$ is a diagonal matrix whose diagonal elements are $\tilde{\lambda}_i = s_i^2$. Eq. A4 states the eigendecomposition of $\mathbf{B}$. Since matrix $\mathbf{B}$ is proportional to the covariance matrix $\mathbf{Q}$ (Eq. A1), we can state the eigendecomposition of the data covariance matrix $\mathbf{Q}$ as

$$\mathbf{Q} = \mathbf{E}\boldsymbol{\Lambda}\mathbf{E}^{\mathrm{T}} \tag{A5}$$

with corresponding eigenvalue problem

$$\mathbf{Q}\mathbf{E} = \boldsymbol{\Lambda}\mathbf{E} \tag{A6}$$

and

$$\boldsymbol{\Lambda} = \frac{1}{n-1}\tilde{\boldsymbol{\Lambda}} \ . \tag{A7}$$

The spatial patterns $e_i$ with size $m \times 1$ are eigenvectors of the data covariance matrix, and the diagonal elements $\lambda_i$ of $\boldsymbol{\Lambda}$ the corresponding eigenvalues $\lambda_i$, which give the amount of variance associated to each $e_i$. From the set of eigenvectors $e_i$ and their corresponding amount of variance $\lambda_i$, the spatial distribution $x_t$ at a given time $t$ can be reconstructed by determination of a suitable amplitude vector $a_t$ (Eq. A3) (see textbooks of linear algebra or statistical data analysis, e.g., Dommenget, 2015).

### A2    Truncation of eigenvalue modes

For practical purposes, reconstruction often uses only the first $l$ leading instead of all $m$ eigenvectors $e_i$ ($1 \le l \le m$), i.e., $\mathbf{E}$, $\boldsymbol{\Lambda}$, and $\mathbf{A}$ are split into a matrix $\mathbf{E}$, $\boldsymbol{\Lambda}$, and $\mathbf{A}$ of size $m \times l$, $l \times l$, and $n \times l$, respectively, which contain the first $l$ leading eigenvector modes, and a matrix $\mathbf{E}'$, $\boldsymbol{\Lambda}'$, and $\mathbf{A}'$, which contain the remaining $m - l$ eigenvector modes. Eqs. A2, A3, and A5 thus become

$$\mathbf{X} = \mathbf{E}\boldsymbol{\Sigma}\mathbf{A}^{\mathrm{T}} + \mathbf{E}'\boldsymbol{\Sigma}'\mathbf{A}'^{\mathrm{T}} \ , \tag{A8}$$

$$x_t = \mathbf{E}\boldsymbol{\Sigma}a_t + \mathbf{E}'\boldsymbol{\Sigma}'a'_t \ , \text{ and} \tag{A9}$$

$$\mathbf{Q} = \mathbf{E}\boldsymbol{\Lambda}\mathbf{E}^{\mathrm{T}} + \mathbf{E}'\boldsymbol{\Lambda}'\mathbf{E}'^{\mathrm{T}} = \mathbf{P} + \mathbf{P}' \ . \tag{A10}$$

The leading eigenvector modes $e_i$ cover most part of the data variance and often represent more "large-scale" spatial patterns, while the higher modes than $l$ contain only a small amount of the data variance and can be seen as "small-scale" spatial pattern





(Lorenz, 1956; Kaplan et al., 2000; Dommenget, 2015). Correspondingly, the first and second term in Eq. A10 give the large

scale and small scale variance $\mathbf{P}$ and $\mathbf{P}'$, respectively, whereas in Eq. A8 and its vector form Eq. A9 they give the large scale and small scale contribution to the spatial data vector $x_t$, respectively.

This split can also be interpreted as decomposition into a "signal" and a "noise" part. Eqs. A8 - A10 with only the $l$ leading modes (i.e., with only the first term) is called a truncated EOF reconstruction, in which dimensionality is reduced from $m$ to $l$ modes. Higher order modes (in $\mathbf{E}'$ and $\mathbf{\Lambda}'$, i.e., the second term) are assumed to be dominated by noise and error and discarded

in a truncated eigenvector or EOF reconstruction (Lorenz, 1956; Kaplan et al., 2000).

### A3   Extrapolation from observational data

To represent a set of $k$ observations at a given time $t$, we use the observational data vector $y_t$ and the observational error $\sigma_t$, both with size $k \times 1$. The transfer operator $\mathbf{H}_t$ is then a "sampling" or observation function that depends on the spatial configuration of the observation points and samples from the data vector $x_t$ of size $m \times 1$ so that the result $\hat{x}_t$,

$$\hat{x}_t = \mathbf{H}_t x_t \ , \tag{A11}$$

has the same size $k \times 1$ as the observations. To be comparable with $\hat{x}_t$, the observational data $y_t$ are expressed as deviation from the space-dependent temporal mean $\overline{x}_m$, too (Elken et al., 2019).

The eigenvalue reconstruction in truncated form (Eq. A9) then becomes:

$$\mathbf{H}_t x_t = \mathbf{H}_t \mathbf{E} \mathbf{\Sigma} a_t \ . \tag{A12}$$

Minimization of a suitable cost function $Q$, e.g., for least-squares optimization as in Elken et al. (2019),

$$Q(\alpha_t) = (\mathbf{H}_t \mathbf{E} \alpha_t - y_t)^{\mathrm{T}} (\mathbf{H}_t \mathbf{E} \alpha_t - y_t) \ , \text{with} \tag{A13}$$

$$\alpha_t = \mathbf{\Sigma} a_t \ , \tag{A14}$$

yields a system of $l$ linear equations with $l$ unknowns (Elken et al., 2019):

$$\mathbf{D} \alpha_t = \mathrm{h} \ , \text{with} \tag{A15}$$

$$\mathbf{D} = \mathbf{E}^{\mathrm{T}} \mathbf{H}_t^{\mathrm{T}} \mathbf{H}_t \mathbf{E} \ , \text{and} \tag{A16}$$

$$\mathrm{h} = \mathbf{E}^{\mathrm{T}} \mathbf{H}_t^{\mathrm{T}} y_t \ , \tag{A17}$$

which provides the observation-based eigenvector amplitudes $\alpha_t$ or $a_t$ (Eq. A14), respectively.

With Eq. A9 in truncated form and Eq. A14, the data vector $x_t$ can be obtained, i.e., an extrapolation from $k \times 1$ observational data points $y_t$ to the entire spatial domain of $m \times 1$ data points $x_t$ with help of the truncated eigenvalue decomposition of $m \times l$

eigenvectors and $l$ eigenvalues $e_i$ and $\lambda_i$, or $\mathbf{E}$ and $\mathbf{\Lambda}$, respectively.

### A4   Error considerations

Both "observational error" and "representational error" impact the determination of the eigenvector amplitudes $a_t$. Observational error $\sigma_t$ includes both instrumental and sampling error and can be used to limit the impact of observational data





constraints $y_k$ (Kaplan et al., 2000). Representational error is the error made by truncation, i.e., by using only the $l$ largest,
leading eigenvectors and neglecting the remainder of the spectrum, which can be derived from the "small-scale" covariance $\mathbf{P}'$
contribution (Kaplan et al., 2000).

The error covariance matrix $\mathcal{R}$ is the sum of the observational data error covariance $\mathbf{S}$ and a term that accounts for the
covariance created in the truncated modes $\mathbf{E}'$ not resolved by the analysis:

$$\mathcal{R} = \mathbf{S} + \mathbf{H}_t \mathbf{E}' \mathbf{\Lambda}' \mathbf{E}'^T \mathbf{H}_t'^T = \mathbf{S} + \mathbf{H}_t \mathbf{P}' \mathbf{H}_t'^T \ , \tag{A18}$$

where $\mathbf{S}$ is a diagonal matrix with the observations' variance $\sigma_t^2$ on the diagonal and $\mathbf{H}_t$ the aforementioned sampling operator
(Kaplan et al., 2000).

By addition of constraints on the cost function $Q$ (a) to limit the estimated eigenvector amplitudes based on the amount of
variance they explain in the original eigenvalue decomposition, and (b) that $a_t$ determination is limited by the uncertainty of
observations and from truncation (see above), Eq. A13 is modified to (Kaplan et al., 2000):

$$Q(\alpha_t) = (\mathbf{H}_t \mathbf{E} \alpha_t - y_t)^{\mathrm{T}} \mathcal{R}^{-1} (\mathbf{H}_t \mathbf{E} \alpha_t - y_t) + \alpha_t^{\mathrm{T}} \mathbf{\Lambda}^{-1} \alpha_t \ , \tag{A19}$$

and the solution Eqs. A15-A17 thus becomes (Kaplan et al., 2000):

$$\mathbf{D} \alpha_t = \mathrm{h} \ , \text{with} \tag{A20}$$

$$\mathbf{D} = \mathbf{E}^T \mathbf{H}_t^{\mathrm{T}} \mathcal{R}^{-1} \mathbf{H}_t \mathbf{E} + \mathbf{\Lambda}^{-1} \ , \text{ and} \tag{A21}$$

$$\mathrm{h} = \mathbf{E}^{\mathrm{T}} \mathbf{H}_t^{\mathrm{T}} \mathcal{R}^{-1} y_t \ . \tag{A22}$$

On the other hand, the large scale portion of the data covariance $\mathbf{Q}$ (Eq. A10) can be obtained from this solution according
to (Kaplan et al., 2000):

$$\mathbf{P} = \mathbf{E} \mathbf{D}^{-1} \mathbf{E}^{\mathrm{T}} \ . \tag{A23}$$

$\mathbf{P}$ can be seen as the mapping uncertainty of the extrapolation.

Note that the calculation of $\mathbf{P}$ only requires the sampling operator $\mathbf{H}_t$, i.e., where samples are present, as well as an observa-
tional error estimate $\mathbf{S}$, but not the actual observations or amplitudes at these locations. In case of absence of any observation,
$\mathbf{H}_t$ contains only zeros and eqs. A21 and A23 then simplify to $\mathbf{D} = \mathbf{\Lambda}^{-1}$ and $\mathbf{P} = \mathbf{E} \mathbf{\Lambda} \mathbf{E}^{\mathrm{T}}$ (compare Eq. A10), respectively.
Addition of observational constraints thus reduces the mapping uncertainty $\mathbf{P}$.

For our purposes, we assume that *off-diagonal* elements in the covariance $\mathbf{P}'$ are negligible, i.e., that the truncated eigenvec-
tor modes $\mathbf{E}'$ are sufficiently small scale and their variances $\mathbf{\Lambda}'$ are small enough that they do not show a noticeable covariance
contribution (Eq. A18). We therefore approximate $\mathbf{P}'$ by a diagonal matrix whose diagonal elements are obtained from rear-
rangement of Eq. A10:

$$\mathrm{diag}(\mathbf{P}') = \mathrm{diag}(\mathbf{Q}) - \mathrm{diag}(\mathbf{E} \mathbf{\Lambda} \mathbf{E}^{\mathrm{T}}) \ , \tag{A24}$$

where $\mathbf{Q}$ is accessible from the data $\mathbf{X}$ (Eq. A1) and the second term from the truncated eigenvalue decomposition.



## A5   Number of significant modes

A critical aspect before any extrapolation from observations is how many modes $e_i$ to use, or – in other words – where
to truncate reconstruction, i.e., how to distinguish between modes representing desired variability and modes representing
"noise". Often, an arbitrary threshold of the total variance covered by the $l$ leading modes is chosen (e.g., Kaplan et al., 2000).

Here, we apply the DINEOF variant of SVD or EOF decomposition of the dataset $\mathbf{X}$ to find the $l_{\max}$ leading eigenvectors
$e_i$. The number $l_{\max}$ of significant modes is determined by a cross-validation procedure from the data $\mathbf{X}$ (Beckers and Rixen,

2003) and we use this $l_{\max}$ as the upper bound for $l$ in our ensemble approach (see below).

## A6   Ensemble approach for robustness and locality

Depending on the spatial distribution or clustering of observations $y_t$, not all amplitudes of the $l_{\max}$ spatial eigenvectors $e_i$
may be well-constrained. In some configurations, e.g., with few or clustered observations, where $\mathbf{H}_t$ selects only few of the
$m$ spatial data points, a reconstruction with fewer modes than $l_{\max}$ may give a solution with a smaller mapping uncertainty $\mathbf{P}$

than the truncated reconstruction with all $l_{\max}$ modes $e_i$.

We therefore use an ensemble approach over the truncated reconstructions where we vary $l$ from just 1 mode to $l_{\max}$ modes,

$$\overline{\mathcal{X}} = \frac{\sum_{l=1}^{l_{\max}} w_l \cdot \mathcal{X}_l}{\sum_{l=1}^{l_{\max}} w_l} \, , \tag{A25}$$

where the weights $w_l$ (with size $m \times 1$) are based on the diagonal elements of $\mathbf{P}$, i.e., the variance $\sigma_{\mathbf{P},t}^2$ or mapping uncertainty
of the extrapolation:

$$w_l = 1 \left/ \frac{\sigma_{\mathbf{P},t}^2}{\sum^m \sigma_{\mathbf{P},t}^2} \right. \, . \tag{A26}$$

As the mapped variance $\mathrm{diag}(\mathbf{P})$ varies in magnitude with the number of eigenvectors included, the weights $w_l$ for a given
number of modes $l$ are normalized by the total variance over all $m$ data points, which gives a normalized spatial weight
vector $w_l$ (Eq. A26). For a given spatial data point $m$, preference is thus given to reconstructions where it is among the better
constrained ones for the given number of modes $l$.

For $\mathcal{X}_l$, each using only the first $l$ ($\leq l_{\max}$) eigenvector modes (Eq. A25), we insert:

– the reconstructed data vector $x_t$ (from Eq. A9 in truncated form) to yield as $\overline{\mathcal{X}}$ the ensemble mean $\overline{x_t}$,

– the term $(\overline{x_t} - x_t)^2$ to yield as $\overline{\mathcal{X}}$ the biased weighted sample variance about the ensemble mean $\sigma_{t,\mathrm{mean}}^2$,

– the mapping variance $\sigma_{\mathbf{P},t}^2$ (i.e., the diagonal of $\mathbf{P}$; Eq. A23) to yield as $\overline{\mathcal{X}}$ the remaining mean "large-scale" variance
$\overline{\sigma_{\mathbf{P},t}^2}$,

– the approximation of the truncated variance $\sigma_{\mathbf{P}',t}^2$ (i.e., the diagonal of $\mathbf{P}'$; Eq. A24) to yield as $\overline{\mathcal{X}}$ the mean "small-scale"
variance $\overline{\sigma_{\mathbf{P}',t}^2}$, and





– the number of modes $l$ itself to yield as $\overline{\mathcal{X}}$ the average number of modes $\bar{l}$ used in the reconstruction

as a spatial vector of size $m \times 1$ each, i.e., on the full spatial domain. The reconstructed data vector $x_{t,\text{reconstr}}$ and the total variance of reconstruction $\sigma^2_{t,\text{reconstr}}$ are given by

$$x_{t,\text{reconstr}} = \overline{x_t} \tag{A27}$$

$$\sigma^2_{t,\text{reconstr}} = \sigma^2_{t,\text{mean}} + \overline{\sigma^2_{\mathbf{P},t}} + \overline{\sigma^2_{\mathbf{P}',t}} \tag{A28}$$

**A7 Temporal coherence**

So far, we considered the observations $y_t$ and the reconstructed data vector $x_t$ to originate and be valid for a given time $t$, respectively. However, in practice, observations $y$ will spread over a certain time extent $\Delta t$, e.g., days for a single SOOP transect to weeks for a basin-covering cruise. If this time extent (e.g., duration of a cruise) is comparable to the time scale of the system's dynamics (e.g., spring surface warming or bloom onset), distortions in the reconstructed fields $x_t$ may occur (Elken et al., 2019) as different "states" of the system could be sampled at different space-time locations.

To collate temporally extended observations into a common synoptic reconstruction without artifacts, Elken et al. (2019) add a linear temporal (short-term) trend for each eigenvector mode to the optimization, thus collating observations made at different observation times $t_p$ to a common reference time instance $t_\circ$.

To this end,

– the time difference $\Delta t_p$ between observation and reference time is introduced:

$$\Delta t_p = t_p - t_\circ . \tag{A29}$$

– The sampling operator $\mathbf{H}_t$ is extended to not only sample from the eigenvector $e_i$ the value, $\chi$, corresponding to the observation, but also the value times the time difference, $\chi \cdot \Delta t_p$, corresponding to a (short-term) time gradient. $\mathbf{H}_t\mathbf{E}$ thus becomes a matrix of size $k \times 2l$.

– The $l$ eigenvalues are replicated with a constant scaling factor, so that $\boldsymbol{\Lambda}$ becomes a $2l \times 2l$ diagonal matrix. I.e., both the value (first $l$ eigenvalues) and its time difference (second $l$ eigenvalues times scaling factor) follow the same eigenvectors $e_i$ order and importance with the scaling factor to determine the balance between between both. Based on preliminary tests, we used an empirical scaling between spatial value pattern and temporal trend pattern of $9 \cdot 10^{-4}$, equal to a ratio of approx. 1 $\mu$atm per 33 days.

– The eigenvalue amplitude vector $a_t$ is doubled in size to $2l \times 1$, where the first $l$ values continue to represent the eigenvectors' $e_i$ amplitudes, while the second $l$ values represent their temporal derivative.

Thus, the system of linear equations (Eqs. A20-A22) becomes a system of $2l$ equations with $2l$ unknowns, and the reconstruction (Eq. A9) provides both an extrapolation to the entire spatial domain of $m \times 1$ data locations at reference time $t_\circ$ as well as a temporal trend of change on the $m \times 1$ data locations at reference time $t_\circ$ (Elken et al., 2019).
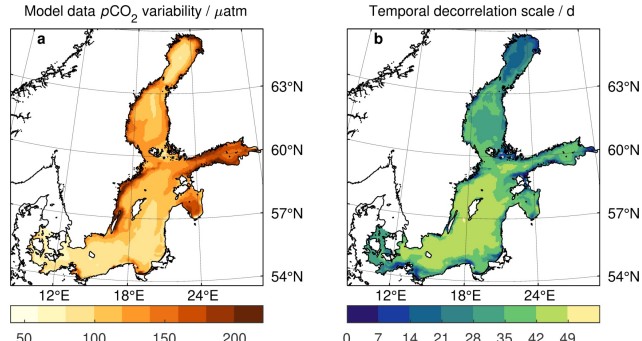

**Figure A1.** Model data $\mathbf{X}$ characteristics: (a) $p\mathrm{CO_2}$ variability shown by the square-root of the model data covariance matrix $\mathbf{Q}$'s diagonal elements, i.e., the standard deviation of the $p\mathrm{CO_2}$ data $\mathbf{X}$ at each location. (b) $p\mathrm{CO_2}$ variability shown by the temporal decorrelation time scale, i.e., the time (in days) at which the lagged autocorrelation drops below a threshold of 0.63 correlation. Less than 1 % of locations have a temporal decorrelation scale of $\leq 7$ d, while about 25 % are within $\leq 30$ d. To adequately reflect the $p\mathrm{CO_2}$ data dynamics in the variability pattern extraction (section 2.2.1), a weekly aggregation of the model data $\mathbf{X}$ was therefore chosen compared to, e.g., daily or monthly model output.

**Appendix B: Seasonal $p\mathrm{CO_2}$ from value and linear trend climatology by cubic Hermite spline calculation**

With a $p\mathrm{CO_2}$ value $x$ and linear $p\mathrm{CO_2}$ trend $d$ given for each month of the climatology, the seasonal evolution of $p\mathrm{CO_2}$ at a given location $m$ can be calculated as a cubic Hermite spline according to:

$$
\begin{aligned}
x_t = {} & h_{00}(\gamma) \cdot x_k \quad + h_{10}(\gamma) \cdot d_k \quad \cdot (t_{k+1} - t_k) + \ldots \\
& h_{01}(\gamma) \cdot x_{k+1} + h_{11}(\gamma) \cdot d_{k+1} \cdot (t_{k+1} - t_k) \ , \text{ with}
\end{aligned}
\tag{B1}
$$

$$h_{00}(\gamma) = 2\gamma^3 - 3\gamma^2 + 1 \ , \tag{B2}$$

$$h_{10}(\gamma) = \ \gamma^3 - 2\gamma^2 + \gamma \ , \tag{B3}$$

$$h_{01}(\gamma) = -2\gamma^3 + 3\gamma^2 \ , \tag{B4}$$

$$h_{11}(\gamma) = \ \gamma^3 - \gamma^2 \ , \text{ and} \tag{B5}$$

$$\gamma = \frac{t - t_k}{t_{k+1} - t_k} \ . \tag{B6}$$

Here, $t_k$ and $t_{k+1}$ are the times of the climatological month before and after the time $t$ of interest, and $\gamma$ gives the normalized time. $h(\gamma)$ are Hermite basis functions and $x_k$ and $x_{k+1}$ as well as $d_k$ and $d_{k+1}$ are the associated monthly $p\mathrm{CO_2}$ value and

$p\mathrm{CO_2}$ trend, respectively, which determine the seasonal $p\mathrm{CO_2}$ $x_t$ at time $t$.

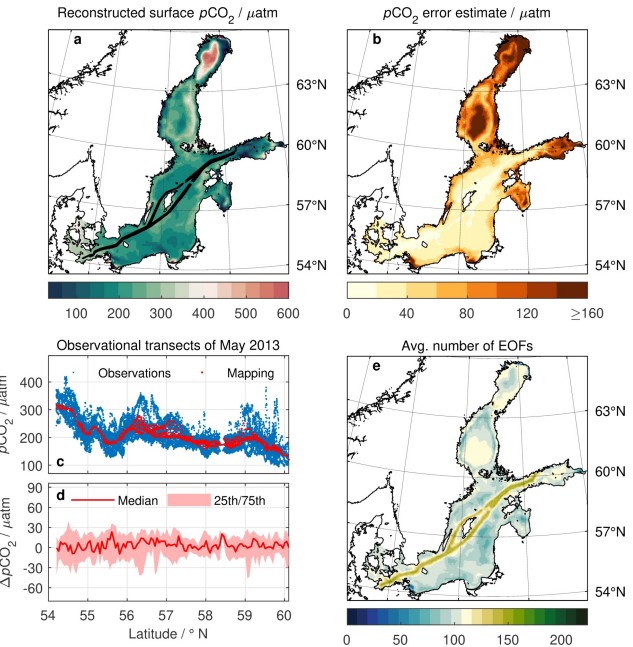

**Figure A2.** Same as Fig. 3 but without temporally collating the observations to the middle of the month (details in section A7). The general picture of (a) reconstructed $pCO_2$ $x_{t,\mathrm{reconstr}}$, (b) $pCO_2$ error estimate $\sigma_{t,\mathrm{reconstr}}$, and (e) average number of patterns $\bar{l}$ of the reconstruction is comparable to the temporally coherent reconstruction (Fig. 3a, b, g, respectively), but (c) mapped $pCO_2$ (red) gives only one uniform value throughout the month and (d) differences to observations are therefore increased (compare Fig. 3e, f).

.

*Author contributions.* HCB and GR conceived the study. The method was developed by HCB with important input by EJ and GR. TN performed the model simulations and HCB the analysis. HCB lead the manuscript writing with contributions by all co-authors.

*Competing interests.* The authors declare that they have no conflict of interest.

*Acknowledgements.* This work was funded by the projects C-SCOPE (grant no. 03F0877D) and BONUS INTEGRAL (grant no. 03F0773A),
which received funding from BONUS (Art. 185), funded jointly by the EU, the German Federal Ministry of Education and Research, the Swedish Research Council Formas, the Academy of Finland, the Polish National Centre for Research and Development, and the Estonian Research Council. Computational power was provided by the North-German Supercomputing Alliance (HLRN). Measurements on SOOP *Finnmaid* were temporarily (2009–2011) funded by the German Federal Ministry of Education and Research in the framework of the BONUS projects Baltic-C (grant no. 03F0486A); Baltic Gas (grant no. 03F0488B); and, since 2012, ICOS-D (grant nos. 01LK1101F
and 01LK1224D). Additional support came from the JERICO-S3 project, funded by the European Commission's H2020 Framework Programme under grant agreement no. 871153. Efforts by A. Willstrand-Wranne (SMHI) and T. Steinhoff (GEOMAR/NORCE) around SOOP



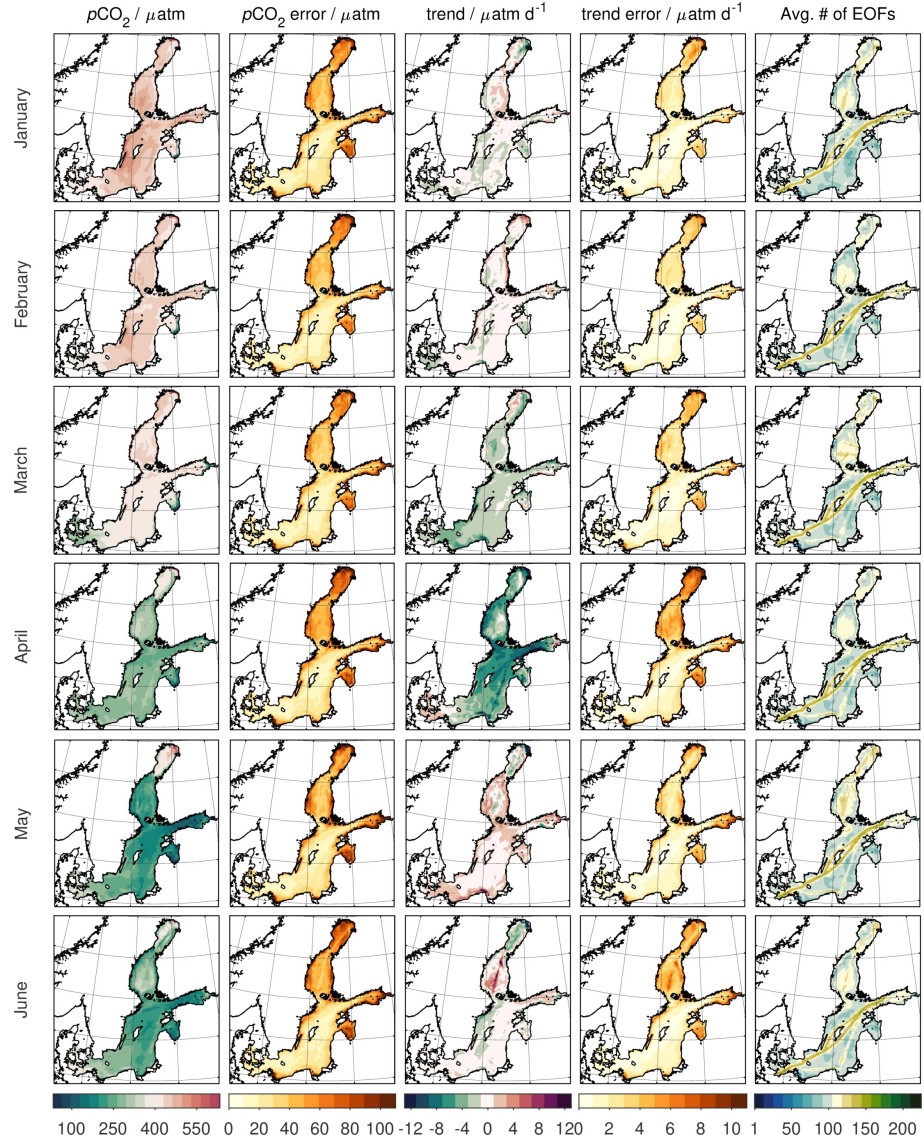

**Figure A3.** Monthly climatology $\mathbf{Y}$ of surface $pCO_2$, $pCO_2$ error estimate, $pCO_2$ trend, $pCO_2$ trend error estimate, and average number of EOF patterns in the ensemble (left to right) for the months Jan. – June (top to bottom).

*Tavastland* observations are highly acknowledged. The Surface Ocean $CO_2$ Atlas (SOCAT) is an international effort, endorsed by the International Ocean Carbon Coordination Project (IOCCP), the Surface Ocean Lower Atmosphere Study (SOLAS) and the Integrated Marine Biosphere Research (IMBeR) program, to deliver a uniformly quality-controlled surface ocean $CO_2$ database. The many researchers and

funding agencies responsible for the collection of data and quality control are thanked for their contributions to SOCAT.

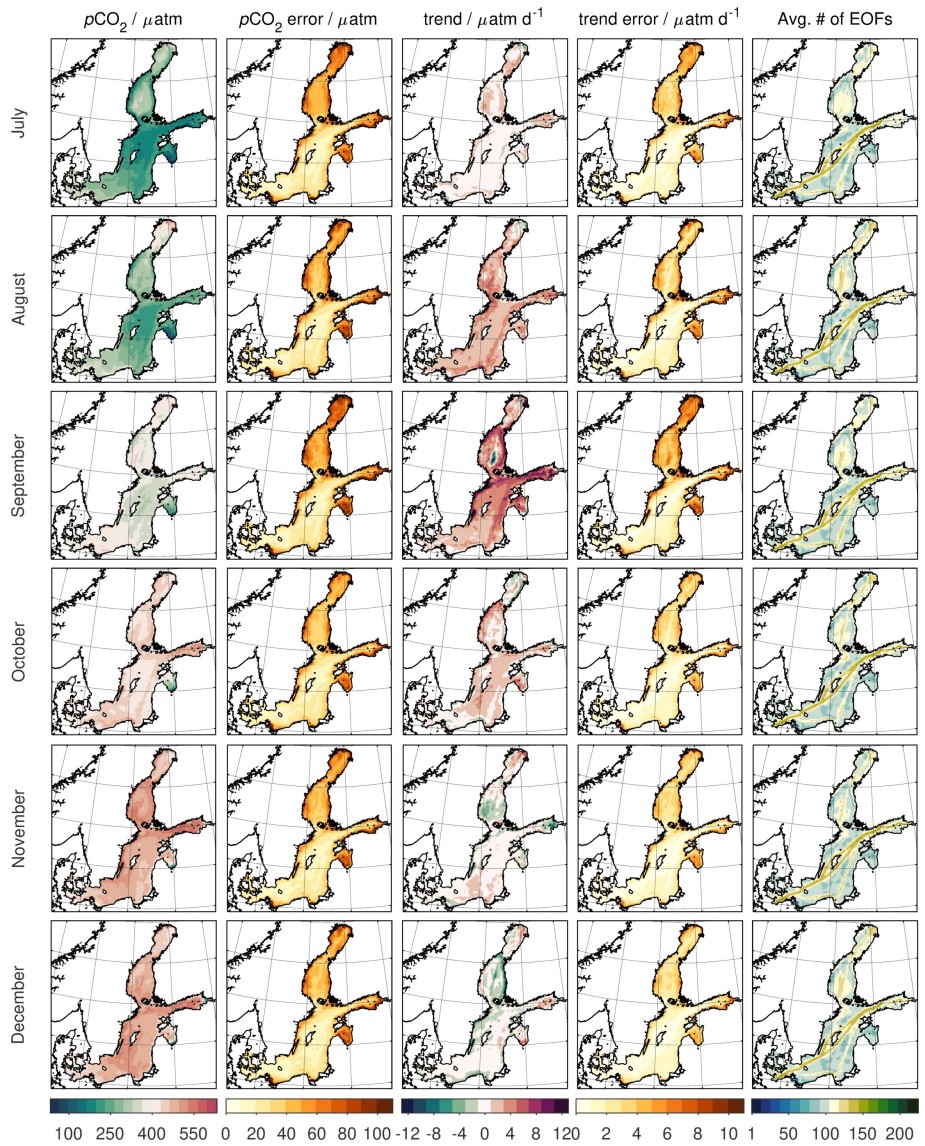

**Figure A4.** Same as Fig. A3 but for July – Dec.

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
