# Peer review of "A regional $pCO_2$ climatology of the Baltic Sea from in situ $pCO_2$ observations and a model-based extrapolation approach"

_Earth System Science Data, 2023_

## Author Comment (AC1)

Answers to referee comments are given in blue.

**Anonymous Referee #1, 07 Sep 2023**

Bittig et al developed a new method to construct a $pCO_2$ climatology in the Baltic Sea. The method is novel, clever and provides a clear and significant improvement towards what currently exists and the results are useful for multiple applications stated in the MS itself. I have a couple of comments below that I would like the authors to consider, which I believe would strengthen the MS. Overall though, I do recommend publication of the MS.

Thank you very much for your assessment and your time to improve the manuscript.

The MS itself currently has some shortcomings that I would like the authors to address:

1. Model comparison: If I follow the methodology correctly, the basis of the method is formed by model $pCO_2$ and its variability, which is then (in my own simple words) corrected by observations.

Yes, you are correct.

As a reader, I would be curious to see a comparison between the actual model output and the $pCO_2$ climatologies obtained. Are they very different?

We want to expand manuscript Figure 4 panels (a) and (b) so that they include the model $pCO_2$ as well as the model $pCO_2$ – SOCAT $pCO_2$ difference, as shown below in Fig. 1:

[Figure]

Fig. 1: Updated manuscript Figure 4 with (a) model $pCO_2$ (blue) and (b) the difference between model $pCO_2$ and observed $pCO_2$ (blue) added.

Model $pCO_2$ data show a distribution that is different from observed $pCO_2$ with a fair $R^2$ of 0.66. Moreover, the differences between model and observations show a much wider range than the reconstructed $pCO_2$ fields.

2. My previous point would also to some sort explain how dependent the method is on the initial model state. If the final $pCO_2$ established through the method (steps 2 and 3 in their Figure 1) is not far away from the original model $pCO_2$, one could conclude (a) that there is not sufficient observations, (b) the model already does a perfect job, so why performing steps 2-3 in Figure 1 or (c) the model is very sensitive to the baseline model $pCO_2$ (step1 in Figure 1).

The initial model $pCO_2$ data is the starting point and thus a pivotal part of the approach. Specifically:

(a) Yes, indeed. If there are not sufficient observations (or even 'none'), the final $pCO_2$ has a tendency to mimic the original model $pCO_2$.

This is because the EOF-based approach uses the "deviation from the space-dependent temporal mean" both for the model data vector x as well as for the observations y (l. 398; 446f). With little or no data constraints, the amplitudes

of the variability patterns (in the ensemble) tend towards zero, i.e., towards zero deviation, which is equivalent to the spatially-resolved temporal mean $pCO_2$ of the model.

(b) If there were no mismatch and if the standard deviation on the differences were similarly small as for the reconstruction, yes. Both is not the case in our setting (but could be elsewhere).

(c) Only partly, namely where there are no observations (see a).

3. The same is true for trends. Are model trends significantly different from the trends obtained in step 4 of Figure 1?

Yes, they are, see Fig. 2 below.

[Figure]

*Fig. 2: Long-term trend G of surface $pCO_2$ from the model and Eq. 1. Note that data coverage in the model is uniform across the model domain, i.e., a higher portion of locations give significant trends compared to manuscript Figure 7 (with, e.g., scarce observations in the Northern basins). The coherent $pCO_2$ increase is linked to the prescribed atmospheric $pCO_2$ increase in the model.*

Given limitations of the pure model and to not distract from the established dataset and its trend, we suggest to make only a brief mention of the different model trend in the text and to not add the figure.

4. The authors in-depth discuss methodological differences with other climatologies (e.g. Becker et al 2021, Parard et al 2016) but there is no actual comparison. As a reader I would be very interested on how this new climatology compares with the state of the art in terms of output

Thank you for your comment. We suggest to add the below Fig. 3 to the appendix, which shows the average $pCO_2$ for one month per season across the different climatologies.

Given that, including our work, there are now a few Baltic Sea $pCO_2$ climatologies that essentially use the same source data (i.e., SOCAT $pCO_2$ observations), we find it important to work out strengths, weaknesses, and limitations of each approach, so that a user can take an informed decision of why going for one or another climatology.

Taking Referee #2's comment into consideration, we will check where section 4.2 can be shortened and will keep a more condensed version in the manuscript as guidance to users.

[Figure]

*Fig. 3: Mean monthly pCO₂ distribution for one month per season (left to right) for different climatologies: Parard et al. 2016 (a-d), Becker et al. 2021 (e-h), and our work (i-l). The first two climatologies are proxy-based, where artefacts of the proxy directly translate to pCO₂, e.g., for winter time satellite retrievals (a, d) or the subtle 5°x4° gridding in Becker et al. (e-h). Our approach uses observed pCO₂ data directly for mapping, which gives fewer artefacts but is only applicable when pCO₂ observations are available.*

5. In a method-heavy publication, it is difficult to not overload the reader in the main text, while still maintaining the required detail for reproducibility. I believe the authors have done a great job offering enough detail in the main text so that the reader can follow, while providing all necessary detail in an appendix.

Thank you very much for your appreciation, which we extend to the handling editor, which proposed to move the heavy details to the Appendix for better readability.

I would have like though a few more lines in the main text on how the error calculation is done.

Noted. We will check lines 71f. to become more explicit/detailed.

6. Evaluation: Lines 177-178 "We therefore stepped back from a quantitative attempt but provide just a qualitative picture (Fig. 5)." I dont understand this statement. From a readers perspective you dont gain more from Figure 5. And - despite all shortcomings in a quantitative assessment - this is a new method that may be used in high profile applications (UN stocktake, CDR activities, extreme event studies) so it behooves the authors well to be as quantitative as possible about their approach and potential errors/uncertainties. I believe the authors should do as many evaluations/comparisons as they can, highlighting obviously the caveats attached to them.

We agree that our phrasing may not transmit our intentions (or better: troubles) properly and therefore elaborate on our thoughts and concerns in more detail below.

They arise from an inability to subset the available data (where one part is used in the method and the remaining part for validation) that is objective and not anecdotal or arbitrary.

For illustration, let's consider a spatial subsetting approach, where the domain is divided into 1° x 1° grid boxes to give a black/white chess board-like grid. Data in every other grid box (e.g., all black 1° x 1° boxes) are used for the method, while data in the boxes in between (e.g., all white 1° x 1° boxes) are used for validation. In analogy to manuscript Figure 4, one can then assess the $pCO_2$ difference between reconstructed and observed $pCO_2$, the $pCO_2$ error estimate, or the ratio between $pCO_2$ difference and $pCO_2$ error estimate. Fig. 4 shows the result of such a 1° x 1° subsetting for one month, May 2019, which has the highest data density.

[Figure]

*Fig. 4: Histograms for May 2019 for: (a) The $pCO_2$ difference between reconstructed and observed $pCO_2$ in 1° x 1° boxes not used for reconstruction. (b) The $pCO_2$ error estimate $\sigma_{reconstr}$ of the reconstruction at the observations. (c) The ratio between $pCO_2$ difference and $pCO_2$ error estimate, where a ratio $\leq 1$ means that the observed $pCO_2$ is within $1 \times \sigma_{reconstr}$ of the mapped $pCO_2$. The colours represent 8 different 1° x 1° subsetting grids only offset in latitude/longitude.*

Fig. 4 gives 8 different realizations, where the 1° x 1° grid is only offset spatially to each other, which impacts $pCO_2$ deviation statistics (mean and standard deviation) to a noticeable extent depending on (arbitrarily picked) starting point (Fig. 4a). A bit more coherent are the ranges of $pCO_2$ error estimates at the observed locations (Fig. 4b), and – encouragingly – the colocation of $pCO_2$ error estimates to observed deviations shows both a similar ratio distribution across realizations as well as to manuscript Figure 4d (or Fig. 1 d).

To have realization-/starting point-dependent deviation statistics does not lend credit to have an objective evaluation design. Further, we can re-do the analysis with 2° x 2° grid boxes (Fig. 5) or 3° x 3° grid boxes (Fig. 6):

[Figure]

*Fig. 5: Same as Fig. 4 but for 2° x 2° subsetting grids, where the colours again represent 8 different, spatially offset grids. $pCO_2$ differences as well as $pCO_2$ error estimates are larger than for 1° x 1°. Their ratio, however, preserves a similar distribution to Fig. 4c (and Fig. 1d).*

[Figure]

*Fig. 6: Same as Fig. 4 but for 3° x 3° subsetting grids, where the colours again represent 8 different, spatially offset grids. $pCO_2$ differences as well as $pCO_2$ error estimates are larger than for 1° x 1° or 2° x 2°. Their ratio, however, preserves a similar distribution to Fig. 4c, 5c (and Fig. 1d).*

From the series of 1° x 1°, 2° x 2°, and 3° x 3° chess grid-based subsettings for May 2019 (Fig. 4-6) one can see that evaluation outcomes are not only realization-dependent, but also design-dependent. The $pCO_2$ deviations and concurrently the $pCO_2$ error estimates become larger with larger spatial scales (as would be expected). Also the variations between different realizations become larger.

We conclude that with a given choice of subsetting, one implicitly chooses what statistical values one wants to get out (e.g., 1° x 1° vs. 3° x 3° with better statistics for the 1° x 1° case). Objectivity is further decreased as the actual realization can noticeably impact the result (e.g., blue example vs. orange examples). If a month with less data coverage were chosen, differences between realizations (i.e., which data end up in the training or validation subset) would become even larger.

We agree with the reviewer that as many quantitative evaluations should be done and presented. Based on the above illustration, however, we do have doubts on any subsetting-based evaluation, as the subsetting design will imprint itself on the outcome, and thus puts in an unneglectable portion of arbitrariness. We therefore strived to limit our evaluations to ones which we'd deem objective and from which we can confidently draw conclusions, thus the initial Lines 177-178 "We therefore stepped back from a quantitative attempt but provide just a qualitative picture (Fig. 5)." statement.

Given this reviewer's comments on little extra knowledge gained by our manuscript Figure 5, we can propose to replace the present manuscript Figure 5 with a combined figure Fig. 4-6 shown here, as well as adding relevant text and descriptions into the second part of section 3.2. We would check with the handling editor to advise whether to proceed with the proposition.

7. A smaller remark on line 16: Maybe state what year or decade the anthropocene started - I had to look this up as I was not aware of it (and it makes a difference whether the ocean absorbed 25% or more)

Noted. We will add this to the text.

**Anonymous Referee #2, 11 Sep 2023**

Review of « *A regional pCO₂ climatology of the Baltic Sea from in situ pCO2 observations and a model-based extrapolation approach*» by Bittig et al. submitted for discussion in ESSD

This manuscript reports a new $pCO_2$ climatology for the Baltic sea which is based on a new extrapolation method of $pCO_2$ observations. The presented dataset (actually only available through a temporary link) ...

> Minting of the dataset doi at Pangaea is now completed so that it can be made available properly.

... is composed of a spatially gridded climatology of the surface water $pCO_2$ of the Baltic sea basin at a resolution of three nautical miles based on data collected between 2003 and 2021. A mean average linear trend of $pCO_2$ over the considered period at the same spatial resolution is also presented.

Theses estimates are based on the conjunct use of two datasets:

- Surface $pCO_2$ measurement values from the SOCAT version 2022
- $pCO_2$ estimates from a model (ERGOM version 1.2) tuned for the Baltic sea.

The interpolation method presented in this manuscript is original. It is based on a EOF decomposition of the model dataset to obtain spatial patterns of $pCO_2$ variability which are then constrained through an optimisation process with the observational values. The strength of the method relies on an ensemble approach which allows an uncertainty estimate for each grid cell.

My general opinion is that this is a very interesting manuscript. It reports an original and robust method to extrapolate data. The manuscript is well written and is supported by proper illustrations. It certainly deserves publication. I have only minor concerns which I hope can improve this overall good manuscript.

> Thank you very much for your assessment and your time to improve the manuscript.

I particularly appreciated the fact that most of the details have been wisely presented in the appendix allowing the reader to have an easy to read main manuscript.

> As noted above, we want to extend our appreciation to the handling editor.

**General comments :**

I regret that the final data product (Climatology and long term trend gridded product) ois not clearly presented. I would appreciate a small section describing the dataset (Format, Size, units, etc).

> This is now available at the Pangaea dataset landing page, which will replace the interim data availability statement of the discussion paper.

The section 4.2 of the discussion (« Comparison with other $pCO_2$ mapping approaches is not necessary ») is interesting to read but it does not give some discussion elements to compare the presented dataset with other climatologies. It is a general discussion on mapping approaches which could be added in the introduction.

> See response to Referee #1 comment 4 above. We find it important to provide some guidance on which Baltic Sea $pCO_2$ climatology (not) to use for what reasons and propose to update this section to make it more concise.

Section 4.3 of the discussion (« Baltic Sea $pCO_2$ climatology ») is discussing the reasons that have led the authors to produce a climatology rather than a monthly gridded product over the entire period. I do not disagree with these arguments but I believe that they could be presented earlier in the manuscript (Section 2 for example).

Good point. We will re-evaluate where the arguments are best placed in the flow of the manuscript.

Section 4.4 of the discussion (« Long-term $p$CO$_2$ trend ») could be simplified by adding a table which would compare the trend in this study to other studies in the Baltic sea.

Agreed and we will add a summary table. We think that some comments on the background of different estimates are warranted for proper interpretation and will keep the extra information in the section 4.4 text.

**Specific comments :**

L26 : What is meant by « smart extrapolation approaches» ? I would suggest just mentioning "extrapolation approaches".

Agreed.

L 150-151 : « We can not observe a tendency of the mapping approach to give extreme values or outliers in absence of observations (compare Fig. 4a). » May be I wrongly understand this sentence but I am not sure to understand how this figure is showing this. This need to be clarified.

Indeed, this is not apparent from Fig. 4a, and we will remove the reference. This sentence was motivated by a bit of a "worst case" consideration: As proxy-based techniques are fully reliant on the quality of the proxies, they have a tendency to provide extreme values/outlies if proxy data are unreliable as "worst case". (This is now getting illustrated by the here presented Fig. 3a,d; to be placed in the appendix). The "worst case" for our mapping is insufficient/absence of data, which equals to a tendency to drive estimates towards the original model mean $p$CO$_2$ rather than extreme values/outlier (see also Referee #1 response 2.a), which is what we wanted to convey here. We will clarify this aspect.

---

## Author Response (AR2)

1     Thank you for your assessment and helpful comments during the review process. The dataset has been published at PANGAEA and the doi minted and added to the data availability statement.